# ZeroSARAH: Efficient Nonconvex Finite-Sum Optimization with Zero Full Gradient Computations

## Abstract

We propose ZeroSARAH—a novel variant of the variance-reduced method SARAH (Nguyen et al., 2017)—for minimizing the average of a large number of nonconvex functions $\frac{1}{n}\sum_{i=1}^{n} f_i(x)$. To the best of our knowledge, in this nonconvex finite-sum regime, all existing variance-reduced methods, including SARAH, SVRG, SAGA and their variants, need to compute the full gradient over all $n$ data samples at the initial point $x^0$, and then periodically compute the full gradient once every few iterations (for SVRG, SARAH and their variants). Note that SVRG, SAGA and their variants typically achieve weaker convergence results than variants of SARAH: $n^{2/3}/\epsilon^2$ vs. $n^{1/2}/\epsilon^2$. Thus we focus on the variant of SARAH. The proposed ZeroSARAH and its distributed variant D-ZeroSARAH are the *first* variance-reduced algorithms which *do not require any full gradient computations*, not even for the initial point. Moreover, for both standard and distributed settings, we show that ZeroSARAH and D-ZeroSARAH obtain new state-of-the-art convergence results, which can improve the previous best-known result (given by e.g., SPIDER, SARAH, and PAGE) in certain regimes. Avoiding any full gradient computations (which are time-consuming steps) is important in many applications as the number of data samples $n$ usually is very large. Especially in the distributed setting, periodic computation of full gradient over all data samples needs to periodically synchronize all clients/devices/machines, which may be impossible or unaffordable. Thus, we expect that ZeroSARAH/D-ZeroSARAH will have a practical impact in distributed and federated learning where full device participation is impractical.

## 1 Introduction

Nonconvex optimization is ubiquitous across many domains of machine learning (Jain & Kar, 2017), especially in training deep neural networks. In this paper, we consider the nonconvex finite-sum problems of the form

$$\min_{x\in\mathbb{R}^d}\left\{f(x) := \frac{1}{n}\sum_{i=1}^{n} f_i(x)\right\}, \tag{1}$$

where $f : \mathbb{R}^d \to \mathbb{R}$ is a differentiable and possibly nonconvex function. Problem (1) captures the standard empirical risk minimization problems in machine learning (Shalev-Shwartz & Ben-David, 2014). There are $n$ data samples and $f_i$ denotes the loss associated with $i$-th data sample. We assume the functions $f_i : \mathbb{R}^d \to \mathbb{R}$ for all $i \in [n] := \{1, 2, \ldots, n\}$ are also differentiable and possibly nonconvex functions.

Beyond the standard/centralized problem (1), we further consider the *distributed/federated* nonconvex problems:

$$\min_{x\in\mathbb{R}^d}\left\{f(x) := \frac{1}{n}\sum_{i=1}^{n} f_i(x)\right\}, \quad f_i(x) := \frac{1}{m}\sum_{j=1}^{m} f_{i,j}(x), \tag{2}$$

where $n$ denotes the number of clients/devices/machines, $f_i$ denotes the loss associated with $m$ data samples stored on client $i$, and all functions are differentiable and can be nonconvex. Avoiding any full gradient computations is important especially in this distributed setting (2), periodic computation of full gradient over all data samples needs to periodically synchronize all clients, which may be impossible or very hard to achieve.

There has been extensive research in designing first-order (gradient-type) methods for solving centralized/distributed nonconvex problems (1) and (2) such as SGD, SVRG, SAGA, SCSG, SARAH and their variants, e.g., (Ghadimi & Lan, 2013; Ghadimi et al., 2016; Allen-Zhu & Hazan, 2016; Reddi et al., 2016; Lei et al., 2017; Li & Li, 2018; Zhou et al., 2018; Fang et al., 2018; Wang et al., 2018; Ge et al., 2019; Pham et al., 2019; Li, 2019; Li & Richtárik, 2020; Horváth et al., 2020; Li et al., 2021). Note that SVRG and SAGA variants typically achieve weaker convergence results than SARAH variants, i.e., $n^{2/3}/\epsilon^2$ vs. $\sqrt{n}/\epsilon^2$. Thus the current best convergence results are achieved by SARAH variants such as SPIDER (Fang et al., 2018), SARAH (Pham et al., 2019) and PAGE (Li et al., 2021; Li, 2021).

However, all of these variance-reduced algorithms (no matter based on SVRG, SAGA or SARAH) require full gradient computations (i.e., compute $\nabla f(x) = \frac{1}{n}\sum_{i=1}^n \nabla f_i(x)$) without assuming additional assumptions except standard $L$-smoothness assumptions. We would like to point out that under an additional bounded variance assumption (e.g., $\mathbb{E}_i[\|\nabla f_i(x) - \nabla f(x)\|^2] \leq \sigma^2, \forall x \in \mathbb{R}^d$), some of them (such as SCSG (Lei et al., 2017), SVRG+ (Li & Li, 2018), PAGE (Li et al., 2021)) may avoid full gradient computations by using a large minibatch of stochastic gradients instead (usually the minibatch size is $O(\sigma^2/\epsilon^2)$). Clearly, there exist some drawbacks: i) $\sigma^2$ usually is not known; ii) if the target error $\epsilon$ is very small (defined as $\mathbb{E}[\|\nabla f(\widehat{x})\|^2] \leq \epsilon^2$ in Definition 1) or $\sigma$ is very large, then the minibatch size $O(\sigma^2/\epsilon^2)$ is still very large for replacing full gradient computations.

In this paper, we only consider algorithms under the standard $L$-smoothness assumptions, without assuming any other additional assumptions (such as bounded variance assumption mentioned above). Thus, all existing variance-reduced methods, including SARAH, SVRG, SAGA and their variants, need to compute the full gradient over all $n$ data samples at the initial point $x^0$, and then periodically compute the full gradient once every few iterations (for SVRG, SARAH and their variants). However, full gradient computations are time-consuming steps in many applications as the number of data samples $n$ usually is very large. Especially in the distributed setting, periodic computation of full gradient needs to periodically synchronize all clients/devices, which usually is impractical. Motivated by this, we focus on designing new algorithms which do not require any full gradient computations for solving standard and distributed nonconvex problems (1)–(2).

## 2 OUR CONTRIBUTIONS

In this paper, we propose the *first* variance-reduced algorithm ZeroSARAH (and also its distributed variant D-ZeroSARAH) *without computing any full gradients* for solving both standard and distributed nonconvex finite-sum problems (1)–(2). Moreover, ZeroSARAH and Distributed D-ZeroSARAH can obtain new state-of-the-art convergence results which improve previous best-known results (given by e.g., SPIDER, SARAH and PAGE) in certain regimes (see Tables 1–2 for the comparison with previous algorithms). ZeroSARAH is formally described in Algorithm 2, which is a variant of SARAH (Nguyen et al., 2017). See Section 4 for more details and comparisons between ZeroSARAH and SARAH. Then, D-ZeroSARAH is formally described in Algorithm 3 of Section 5, which is a distributed variant of our ZeroSARAH.

Now, we highlight the following results achieved by ZeroSARAH and D-ZeroSARAH:

• ZeroSARAH and D-ZeroSARAH are the *first* variance-reduced algorithms which *do not require any full gradient computations*, not even for the initial point (see Algorithms 2–3 or Tables 1–2). Avoiding any full gradient computations is important in many applications as the number of data samples $n$ usually is very large. Especially in the distributed setting, periodic computation of full gradient over all data samples stored in all clients/devices may be impossible or very hard to achieve. We expect that ZeroSARAH/D-ZeroSARAH will have a practical impact in distributed and federated learning where full device participation is impractical.

• Moreover, ZeroSARAH can recover the previous best-known convergence result $O(n + \frac{\sqrt{n}L\Delta_0}{\epsilon^2})$ (see Table 1 or Corollary 1), and also provide new state-of-the-art convergence results without any

Table 1: Stochastic gradient complexity for finding an $\epsilon$-approximate solution of nonconvex problems (1), under Assumption 1

| Algorithms | Stochastic gradient complexity | Full gradient computation |
|---|---|---|
| GD (Nesterov, 2004) | $O(\frac{nL\Delta_0}{\epsilon^2})$ | Computed for every iteration |
| SVRG (Reddi et al., 2016; Allen-Zhu & Hazan, 2016), SCSG (Lei et al., 2017), SVRG+ (Li & Li, 2018) | $O\left(n + \frac{n^{2/3}L\Delta_0}{\epsilon^2}\right)$ | Computed for the initial point and periodically computed for every $l$ iterations |
| SNVRG (Zhou et al., 2018), Geom-SARAH (Horváth et al., 2020) | $\widetilde{O}\left(n + \frac{\sqrt{n}L\Delta_0}{\epsilon^2}\right)$ | Computed for the initial point and periodically computed for every $l$ iterations |
| SPIDER (Fang et al., 2018), SpiderBoost (Wang et al., 2018), SARAH (Pham et al., 2019), SSRGD (Li, 2019), PAGE (Li et al., 2021) | $O\left(n + \frac{\sqrt{n}L\Delta_0}{\epsilon^2}\right)$ | Computed for the initial point and periodically computed for every $l$ iterations |
| ZeroSARAH (this paper, Corollary 1) | $O\left(n + \frac{\sqrt{n}L\Delta_0}{\epsilon^2}\right)$ | Only computed once for the initial point [1] |
| ZeroSARAH (this paper, Corollary 2) | $O\left(\frac{\sqrt{n}(L\Delta_0 + G_0)}{\epsilon^2}\right)$ | Never computed [2] |

[1] In Corollary 1, ZeroSARAH only computes the full gradient $\nabla f(x^0) = \frac{1}{n}\sum_{i=1}^{n}\nabla f_i(x^0)$ once for the initial point $x^0$, i.e., minibatch size $b_0 = n$, and then $b_k \equiv \sqrt{n}$ for all iterations $k \geq 1$ in Algorithm 2.

[2] In Corollary 2, ZeroSARAH never computes full gradients, i.e., minibatch size $b_k \equiv \sqrt{n}$ for all iterations $k \geq 0$.

full gradient computations (see Table 1 or Corollary 2) which can improve the previous best result in certain regimes.

• Besides, for the distributed nonconvex setting (2), the distributed D-ZeroSARAH (Algorithm 3) enjoys similar benefits as our ZeroSARAH, i.e., D-ZeroSARAH does not need to periodically synchronize all $n$ clients to compute any full gradients, and also provides new state-of-the-art convergence results. See Table 2 and Section 5 for more details.

• Finally, the experimental results in Section 6 show that ZeroSARAH is slightly better than the previous state-of-the-art SARAH. However, we should point out that ZeroSARAH does not compute any full gradients while SARAH needs to periodically compute the full gradients for every $l$ iterations (here $l = \sqrt{n}$). Thus the experiments validate our theoretical results (can be slightly better than SARAH (see Table 1)) and confirm the practical superiority of ZeroSARAH (avoid any full gradient computations). Similar experimental results of D-ZeroSARAH for the distributed setting are provided in Appendix A.2.

## 3 PRELIMINARIES

**Notation:** Let $[n]$ denote the set $\{1, 2, \cdots, n\}$ and $\|\cdot\|$ denote the Euclidean norm for a vector and the spectral norm for a matrix. Let $\langle u, v \rangle$ denote the inner product of two vectors $u$ and $v$. We use $O(\cdot)$ and $\Omega(\cdot)$ to hide the absolute constant, and $\widetilde{O}(\cdot)$ to hide the logarithmic factor. We will write $\Delta_0 := f(x^0) - f^*$, $f^* := \min_{x \in \mathbb{R}^d} f(x)$, $G_0 := \frac{1}{n}\sum_{i=1}^{n}\|\nabla f_i(x^0)\|^2$, $\widehat{\Delta}_0 := f(x^0) - \widehat{f}^*$, $\widehat{f}^* := \frac{1}{n}\sum_{i=1}^{n}\min_{x \in \mathbb{R}^d} f_i(x)$ and $G_0' := \frac{1}{nm}\sum_{i,j=1,1}^{n,m}\|\nabla f_{i,j}(x^0)\|^2$.

Table 2: Stochastic gradient complexity for finding an $\epsilon$-approximate solution of *distributed* nonconvex problems (2), under Assumption 2

| Algorithms | Stochastic gradient complexity | Full gradient computation |
|---|---|---|
| DC-GD [1] (Khaled & Richtárik, 2020; Li & Richtárik, 2020) | $O(\frac{mL\Delta_0}{\epsilon^2})$ | Computed for every iteration |
| D-SARAH [2] (Cen et al., 2020) | $O\left(m + \frac{\sqrt{m \log m}L\Delta_0}{\epsilon^2}\right)$ | Computed for the initial point and periodically computed across all $n$ clients |
| D-GET [2] (Sun et al., 2020) | $O\left(m + \frac{\sqrt{m}L\Delta_0}{\epsilon^2}\right)$ | Computed for the initial point and periodically computed across all $n$ clients |
| SCAFFOLD [3] (Karimireddy et al., 2020) | $O\left(m + \frac{m}{n^{1/3}}\frac{L\Delta_0}{\epsilon^2}\right)$ | Only computed once for the initial point |
| DC-LSVRG/DC-SAGA [1] (Li & Richtárik, 2020) | $O\left(m + \frac{m^{2/3}}{n^{1/3}}\frac{L\Delta_0}{\epsilon^2}\right)$ | Computed for the initial point and periodically computed across all $n$ clients |
| FedPAGE [3] (Zhao et al., 2021) | $O\left(m + \frac{m}{\sqrt{n}}\frac{L\Delta_0}{\epsilon^2}\right)$ | Computed for the initial point and periodically computed across all $n$ clients |
| (Distributed) SARAH/SPIDER/SSRGD [4] (Nguyen et al., 2017; Fang et al., 2018; Li, 2019) | $O\left(m + \sqrt{\frac{m}{n}}\frac{L\Delta_0}{\epsilon^2}\right)$ | Computed for the initial point and periodically computed across all $n$ clients |
| D-ZeroSARAH (this paper, Corollary 3) | $O\left(m + \sqrt{\frac{m}{n}}\frac{L\Delta_0}{\epsilon^2}\right)$ | Only computed once for the initial point |
| D-ZeroSARAH (this paper, Corollary 4) | $O\left(\sqrt{\frac{m}{n}}\frac{L\Delta_0 + G_0'}{\epsilon^2}\right)$ | Never computed |

[1] Distributed compressed methods. Here we translate their results to this distributed setting (2).
[2] Decentralized methods. Here we translate their results to this distributed setting (2).
[3] Federated local methods. Here we translate their results to this distributed setting (2).
[4] Distributed version of previous SARAH-type methods (see e.g., Algorithm 4 in Appendix A.2).

**Definition 1** *A point $\widehat{x}$ is called an $\epsilon$-approximate solution for nonconvex problems* (1) *and* (2) *if* $\mathbb{E}[\|\nabla f(\widehat{x})\|^2] \leq \epsilon^2$.

To show the convergence results, we assume the following standard smoothness assumption for nonconvex problems (1).

**Assumption 1 ($L$-smoothness)** *A function $f_i : \mathbb{R}^d \to \mathbb{R}$ is $L$-smooth if $\exists L > 0$, such that*

$$\|\nabla f_i(x) - \nabla f_i(y)\| \leq L\|x - y\|, \quad \forall x, y \in \mathbb{R}^d. \tag{3}$$

It is easy to see that $f(x) = \frac{1}{n}\sum_{i=1}^{n} f_i(x)$ is also $L$-smooth under Assumption 1. We can also relax Assumption 1 by defining $L_i$-smoothness for each $f_i$. Then if we further define the average $L^2 := \frac{1}{n}\sum_{i=1}^{n} L_i^2$, we know that $f(x) = \frac{1}{n}\sum_{i=1}^{n} f_i(x)$ is also $L$-smooth. Here we use the same $L$ just for simple representation.

For the distributed nonconvex problems (2), we use the following Assumption 2 instead of Assumption 1. Similarly, we can also relax it by defining $L_{i,j}$-smoothness for different $f_{i,j}$. Here we use the same $L$ just for simple representation.

**Assumption 2 ($L$-smoothness)** *A function $f_{i,j} : \mathbb{R}^d \to \mathbb{R}$ is $L$-smooth if $\exists L > 0$, such that*

$$\|\nabla f_{i,j}(x) - \nabla f_{i,j}(y)\| \leq L\|x - y\|, \quad \forall x, y \in \mathbb{R}^d. \tag{4}$$

---

**Algorithm 1** SARAH (Nguyen et al., 2017; Pham et al., 2019)

---

**Input:** initial point $x^0$, epoch length $l$, stepsize $\eta$, minibatch size $b$
1: $\widetilde{x} = x^0$
2: **for** $s = 0, 1, 2, \ldots$ **do**
3:    $x^0 = \widetilde{x}$
4:    $v^0 = \nabla f(x^0) = \frac{1}{n} \sum_{i=1}^{n} \nabla f_i(x^0)$      // compute the full gradient once for every $l$ iterations
5:    $x^1 = x^0 - \eta v^0$
6:    **for** $k = 1, 2, \ldots, l$ **do**
7:       Randomly sample a minibatch data samples $I_b$ with $|I_b| = b$
8:       $v^k = \frac{1}{b} \sum_{i \in I_b} \left( \nabla f_i(x^k) - \nabla f_i(x^{k-1}) \right) + v^{k-1}$
9:       $x^{k+1} = x^k - \eta v^k$
10:   **end for**
11:   $\widetilde{x}$ randomly chosen from $\{x^k\}_{k \in [l]}$ or $\widetilde{x} = x^{l+1}$
12: **end for**

---

**Algorithm 2** SARAH without full gradient computations (ZeroSARAH)

---

**Input:** initial point $x^0$, stepsize $\{\eta_k\}$, minibatch size $\{b_k\}$, parameter $\{\lambda_k\}$
1: $x^{-1} = x^0$
2: $v^{-1} = 0, y_1^{-1} = y_2^{-1} = \cdots = y_n^{-1} = 0$     // no full gradient computation
3: **for** $k = 0, 1, 2, \ldots$ **do**
4:    Randomly sample a minibatch data samples $I_b^k$ with $|I_b^k| = b_k$
5:    $v^k = \frac{1}{b_k} \sum_{i \in I_b^k} \left( \nabla f_i(x^k) - \nabla f_i(x^{k-1}) \right) + (1 - \lambda_k) v^{k-1} + \lambda_k \left( \frac{1}{b_k} \sum_{i \in I_b^k} \left( \nabla f_i(x^{k-1}) - y_i^{k-1} \right) + \frac{1}{n} \sum_{j=1}^{n} y_j^{k-1} \right)$
      // no full gradient computations for $v^k$s
6:    $x^{k+1} = x^k - \eta_k v^k$
7:    $y_i^k = \begin{cases} \nabla f_i(x^k) & \text{for } i \in I_b^k \\ y_i^{k-1} & \text{for } i \notin I_b^k \end{cases}$
      // the update of $\{y_i^k\}$ directly follows from the stochastic gradients computed in Line 5
8: **end for**

---

## 4 ZeroSARAH Algorithm and Its Convergence Results

In this section, we consider the standard/centralized nonconvex problems (1). The distributed setting (2) is considered in the following Section 5.

### 4.1 ZeroSARAH algorithm

We first describe the proposed ZeroSARAH in Algorithm 2, which is a variant of SARAH (Nguyen et al., 2017). To better compare with SARAH and ZeroSARAH, we also recall the original SARAH in Algorithm 1.

Now, we highlight some points for the difference between SARAH and our ZeroSARAH:

• SARAH requires the full gradient computations for every epoch (see Line 4 of Algorithm 1). However, ZeroSARAH combines the past gradient estimator $v^{k-1}$ with another estimator to avoid periodically computing the full gradient. See the difference between Line 8 of Algorithm 1 and Line 5 of Algorithm 2 (also highlighted with blue color).

• The gradient estimator $v^k$ in ZeroSARAH (Line 5 of Algorithm 2) does not require more stochastic gradient computations compared with $v^k$ in SARAH (Line 8 of Algorithm 1) if the minibatch size $b_k = b$.

• The new gradient estimator $v^k$ of ZeroSARAH also leads to simpler algorithmic structure, i.e., single-loop in ZeroSARAH vs. double-loop in SARAH.

• Moreover, the difference of gradient estimator $v^k$ also leads to different results in expectation, i.e., 1) for SARAH: $\mathbb{E}_k[v^k - \nabla f(x^k)] = v^{k-1} - \nabla f(x^{k-1})$; 2) for ZeroSARAH: $\mathbb{E}_k[v^k - \nabla f(x^k)] = (1 - \lambda_k)(v^{k-1} - \nabla f(x^{k-1}))$.

## 4.2 CONVERGENCE RESULTS FOR ZeroSARAH

Now, we present the main convergence theorem (Theorem 1) of ZeroSARAH (Algorithm 2) for solving nonconvex finite-sum problems (1). Subsequently, we formulate two corollaries which present the detailed convergence results by specifying the choice of parameters. In particular, we list the results of these two Corollaries 1–2 in Table 1 for comparing with convergence results of previous works.

**Theorem 1** *Suppose that Assumption 1 holds. Choose stepsize $\eta_k \leq \frac{1}{L\left(1 + \sqrt{M_{k+1}}\right)}$ for any $k \geq 0$, where $M_{k+1} := \frac{2}{\lambda_{k+1} b_{k+1}} + \frac{8\lambda_{k+1} n^2}{b_{k+1}^3}$. Moreover, let $\lambda_0 = 1$, $\gamma_0 \geq \frac{\eta_0}{2\lambda_1}$ and $\alpha_0 \geq \frac{2n\lambda_1 \eta_0}{b_1^2}$. Then the following equation holds for ZeroSARAH (Algorithm 2) for solving problem (1), for any iteration $K \geq 0$:*

$$\mathbb{E}[\|\nabla f(\widehat{x}^K)\|^2] \leq \frac{2\Delta_0}{\sum_{k=0}^{K-1} \eta_k} + \frac{(n - b_0)(4\gamma_0 + 2\alpha_0 b_0)G_0}{nb_0 \sum_{k=0}^{K-1} \eta_k}. \tag{5}$$

**Remark:** Note that we can upper bound both terms on the right-hand side of (5). It means that there is no convergence neighborhood of ZeroSARAH and hence, ZeroSARAH can find an $\epsilon$-approximate solution for any $\epsilon > 0$.

In the following, we provide two detailed convergence results in Corollaries 1 and 2 by specifying two kinds of parameter settings. Note that the algorithm computes full gradient in iteration $k$ if the minibatch $b_k = n$. Our convergence results show that without computing any full gradients actually does not hurt the convergence performance of algorithms (see Table 1).

In particular, we note that the second term of (5) will be deleted if we choose minibatch size $b_0 = n$ for the initial point $x^0$ (see Corollary 1 for more details). Here Corollary 1 only needs to compute the full gradient once for the initialization, and does not compute any full gradients later (i.e., $b_k \equiv \sqrt{n}$ for all $k > 0$).

Also note that even if we choose $b_0 < n$, we can also upper bound the second term of (5). It means that ZeroSARAH can find an $\epsilon$-approximate solution without computing any full gradients even for the initial point, i.e., minibatch size $b_k < n$ for all iterations $k \geq 0$. For instance, we choose $b_k \equiv \sqrt{n}$ for all $k \geq 0$ in Corollary 2 , i.e., ZeroSARAH never computes any full gradients even for the initial point.

**Corollary 1** *Suppose that Assumption 1 holds. Choose stepsize $\eta_k \leq \frac{1}{(1+\sqrt{8})L}$ for any $k \geq 0$, minibatch size $b_k \equiv \sqrt{n}$ and parameter $\lambda_k = \frac{b_k}{2n}$ for any $k \geq 1$. Moreover, let $b_0 = n$ and $\lambda_0 = 1$. Then ZeroSARAH (Algorithm 2) can find an $\epsilon$-approximate solution for problem (1) such that*

$$\mathbb{E}[\|\nabla f(\widehat{x}^K)\|^2] \leq \epsilon^2$$

*and the number of stochastic gradient computations can be bounded by*

$$\#\mathrm{grad} := \sum_{k=0}^{K-1} b_k \leq n + \frac{2(1+\sqrt{8})\sqrt{n}L\Delta_0}{\epsilon^2} = O\Big(n + \frac{\sqrt{n}L\Delta_0}{\epsilon^2}\Big).$$

**Remark:** In Corollary 1, ZeroSARAH only computes the full gradient $\nabla f(x^0) = \frac{1}{n}\sum_{i=1}^n \nabla f_i(x^0)$ once for the initial point $x^0$, i.e., minibatch size $b_0 = n$, and then $b_k \equiv \sqrt{n}$ for all iterations $k \geq 1$ in Algorithm 2.

In the following Corollary 2, we show that ZeroSARAH without computing any full gradients even for the initial point does not hurt its convergence performance.

**Corollary 2** *Suppose that Assumption 1 holds. Choose stepsize $\eta_k \leq \frac{1}{(1+\sqrt{8})L}$ for any $k \geq 0$, minibatch size $b_k \equiv \sqrt{n}$ for any $k \geq 0$, and parameter $\lambda_0 = 1$ and $\lambda_k = \frac{b_k}{2n}$ for any $k \geq 1$. Then ZeroSARAH (Algorithm 2) can find an $\epsilon$-approximate solution for problem* (1) *such that*

$$\mathbb{E}[\|\nabla f(\widehat{x}^K)\|^2] \leq \epsilon^2$$

*and the number of stochastic gradient computations can be bounded by*

$$\#\mathrm{grad} = O\left(\frac{\sqrt{n}(L\Delta_0 + G_0)}{\epsilon^2}\right).$$

*Note that $G_0$ can be bounded by $G_0 \leq 2L\widehat{\Delta}_0$ via L-smoothness Assumption 1, then we also have*

$$\#\mathrm{grad} = O\left(\frac{\sqrt{n}(L\Delta_0 + L\widehat{\Delta}_0)}{\epsilon^2}\right).$$

**Remark:** In Corollary 2, ZeroSARAH never computes any full gradients even for the initial point, i.e., minibatch size $b_k \equiv \sqrt{n}$ for all iterations $k \geq 0$ in Algorithm 2. If we consider $L$, $\Delta_0$, $G_0$ or $\widehat{\Delta}_0$ as constant values then the stochastic gradient complexity in Corollary 2 is $\#\mathrm{grad} = O(\frac{\sqrt{n}}{\epsilon^2})$, i.e., full gradient computations do not appear in ZeroSARAH (Algorithm 2) and the term '$n$' also does not appear in its convergence result. Also note that the parameter settings (i.e., $\{\eta_k\}$, $\{b_k\}$ and $\{\lambda_k\}$ in Algorithm 2) of Corollaries 1 and 2 are exactly the same except for $b_0 = n$ (in Corollary 1) and $b_0 = \sqrt{n}$ (in Corollary 2). Moreover, the parameter settings (i.e., $\{\eta_k\}$, $\{b_k\}$ and $\{\lambda_k\}$) for Corollaries 1 and 2 only require the values of $L$ and $n$, which is the same as all previous algorithms. If one further allows other values, e.g., $\epsilon$, $G_0$ or $\widehat{\Delta}_0$, for setting the initial $b_0$, then the gradient complexity can be further improved (see Appendix D for more details).

## 5 D-ZeroSARAH Algorithm and Its Convergence Results

Now, we consider the *distributed* nonconvex problems (2), i.e., $\min_{x \in \mathbb{R}^d} \left\{ f(x) := \frac{1}{n} \sum_{i=1}^n f_i(x) \right\}$ with $f_i(x) := \frac{1}{m} \sum_{j=1}^m f_{i,j}(x)$, where $n$ denotes the number of clients/devices/machines, $f_i$ denotes the loss associated with $m$ data samples stored on client $i$.

### 5.1 D-ZeroSARAH algorithm

To solve distributed nonconvex problems (2), we propose a distributed variant of ZeroSARAH (called D-ZeroSARAH) and describe it in Algorithm 3. Same as our ZeroSARAH, D-ZeroSARAH also does not need to compute any full gradients at all. Avoiding any full gradient computations is important especially in this distributed setting, periodic computation of full gradient across all $n$ clients may be impossible or unaffordable. Thus, we expect the proposed D-ZeroSARAH (Algorithm 3) will have a practical impact in distributed and federated learning where full device participation is impractical.

### 5.2 Convergence results for D-ZeroSARAH

Similar to ZeroSARAH in Section 4.2, we also first present the main convergence theorem (Theorem 2) of D-ZeroSARAH (Algorithm 3) for solving distributed nonconvex problems (2). Subsequently, we formulate two corollaries which present the detailed convergence results by specifying the choice of parameters. In particular, we list the results of these two Corollaries 3–4 in Table 2 for comparing with convergence results of previous works. Note that here we use the smoothness Assumption 2 instead of Assumption 1 for this distributed setting (2).

**Theorem 2** *Suppose that Assumption 2 holds. Choose stepsize $\eta_k \leq \frac{1}{L\left(1+\sqrt{W_{k+1}}\right)}$ for any $k \geq 0$, where $W_{k+1} := \frac{2}{\lambda_{k+1} s_{k+1} b_{k+1}} + \frac{8\lambda_{k+1} n^2 m^2}{s_{k+1}^3 b_{k+1}^3}$. Moreover, let $\lambda_0 = 1$ and $\theta_0 := \frac{nm}{(nm-1)\lambda_1} + \frac{4nm\lambda_1 s_0 b_0}{s_1^2 b_1^2}$. Then the following equation holds for D-ZeroSARAH (Algorithm 3) for solving distributed problem* (2), *for any iteration $K \geq 0$:*

$$\mathbb{E}[\|\nabla f(\widehat{x}^K)\|^2] \leq \frac{2\Delta_0}{\sum_{k=0}^{K-1} \eta_k} + \frac{(nm - s_0 b_0)\eta_0 \theta_0 G_0'}{nm s_0 b_0 \sum_{k=0}^{K-1} \eta_k}. \tag{6}$$

---

**Algorithm 3** Distributed ZeroSARAH (D-ZeroSARAH)

---

**Input:** initial point $x^0$, parameters $\{\eta_k\}, \{s_k\}, \{b_k\}, \{\lambda_k\}$

1: $x^{-1} = x^0$

2: $v^{-1} = 0, y_1^{-1} = y_2^{-1} = \cdots = y_n^{-1} = 0$      // no full gradient computation

3: **for** $k = 0, 1, 2, \ldots$ **do**

4:     Randomly sample a subset of clients $S^k$ from $n$ clients with size $|S^k| = s_k$

5:     **for** each client $i \in S^k$ **do**

6:        Sample the data minibatch $I_{b_i}^k$ (with size $|I_{b_i}^k| = b_k$) from the $m$ data samples in client $i$

7:        Compute its local minibatch gradient information:

$$g_{i,\text{curr}}^k = \frac{1}{b_k} \sum_{j \in I_{b_i}^k} \nabla f_{i,j}(x^k), \quad g_{i,\text{prev}}^k = \frac{1}{b_k} \sum_{j \in I_{b_i}^k} \nabla f_{i,j}(x^{k-1}), \quad y_{i,\text{prev}}^k = \frac{1}{b_k} \sum_{j \in I_{b_i}^k} y_{i,j}^{k-1}$$

$$y_{i,j}^k = \begin{cases} \nabla f_{i,j}(x^k) & \text{for } j \in I_{b_i}^k \\ y_{i,j}^{k-1} & \text{for } j \notin I_{b_i}^k \end{cases}, \quad y_i^k = \frac{1}{m} \sum_{j=1}^m y_{i,j}^k$$

8:     **end for**

9:     $v^k = \frac{1}{s_k} \sum_{i \in S^k} \left( g_{i,\text{curr}}^k - g_{i,\text{prev}}^k \right) + (1 - \lambda_k) v^{k-1} + \lambda_k \frac{1}{s_k} \sum_{i \in S^k} \left( g_{i,\text{prev}}^k - y_{i,\text{prev}}^k \right) + \lambda_k y^{k-1}$

      // no full gradient computations for $v^k$s

10:    $x^{k+1} = x^k - \eta_k v^k$

11:    $y^k = \frac{1}{n} \sum_{i=1}^n y_i^k$      // here $y_i^k = y_i^{k-1}$ for client $i \notin S^k$

12: **end for**

---

**Corollary 3** *Suppose that Assumption 2 holds. Choose stepsize $\eta_k \leq \frac{1}{(1+\sqrt{8})L}$ for any $k \geq 0$, clients subset size $s_k \equiv \sqrt{n}$, minibatch size $b_k \equiv \sqrt{m}$ and parameter $\lambda_k = \frac{s_k b_k}{2nm}$ for any $k \geq 1$. Moreover, let $s_0 = n$, $b_0 = m$, and $\lambda_0 = 1$. Then* D-ZeroSARAH *(Algorithm 3) can find an $\epsilon$-approximate solution for distributed problem* (2) *such that*

$$\mathbb{E}[\|\nabla f(\widehat{x}^K)\|^2] \leq \epsilon^2$$

*and the number of stochastic gradient computations for each client can be bounded by*

$$\#\text{grad} = O\left( m + \sqrt{\frac{m}{n}} \frac{L\Delta_0}{\epsilon^2} \right).$$

**Corollary 4** *Suppose that Assumption 2 holds. Choose stepsize $\eta_k \leq \frac{1}{(1+\sqrt{8})L}$ for any $k \geq 0$, clients subset size $s_k \equiv \sqrt{n}$ and minibatch size $b_k \equiv \sqrt{m}$ for any $k \geq 0$, and parameter $\lambda_0 = 1$ and $\lambda_k = \frac{s_k b_k}{2nm}$ for any $k \geq 1$. Then* D-ZeroSARAH *(Algorithm 3) can find an $\epsilon$-approximate solution for distributed problem* (2) *such that*

$$\mathbb{E}[\|\nabla f(\widehat{x}^K)\|^2] \leq \epsilon^2$$

*and the number of stochastic gradient computations for each client can be bounded by*

$$\#\text{grad} = O\left( \sqrt{\frac{m}{n}} \frac{L\Delta_0 + G_0'}{\epsilon^2} \right).$$

**Remark:** Similar discussions and remarks of Theorem 1 and Corollaries 1–2 for ZeroSARAH in Section 4.2 also hold for the results of D-ZeroSARAH (i.e., Theorem 2 and Corollaries 3–4).

## 6 EXPERIMENTS

Now, we present the numerical experiments for comparing the performance of our ZeroSARAH/D-ZeroSARAH with previous algorithms. In the experiments, we consider the nonconvex robust linear regression and binary classification with two-layer neural networks, which are used in (Wang et al., 2018; Zhao et al., 2010; Tran-Dinh et al., 2019). All datasets used in our experiments are downloaded from LIBSVM (Chang & Lin, 2011). The detailed description of these objective functions and datasets are provided in Appendix A.1.

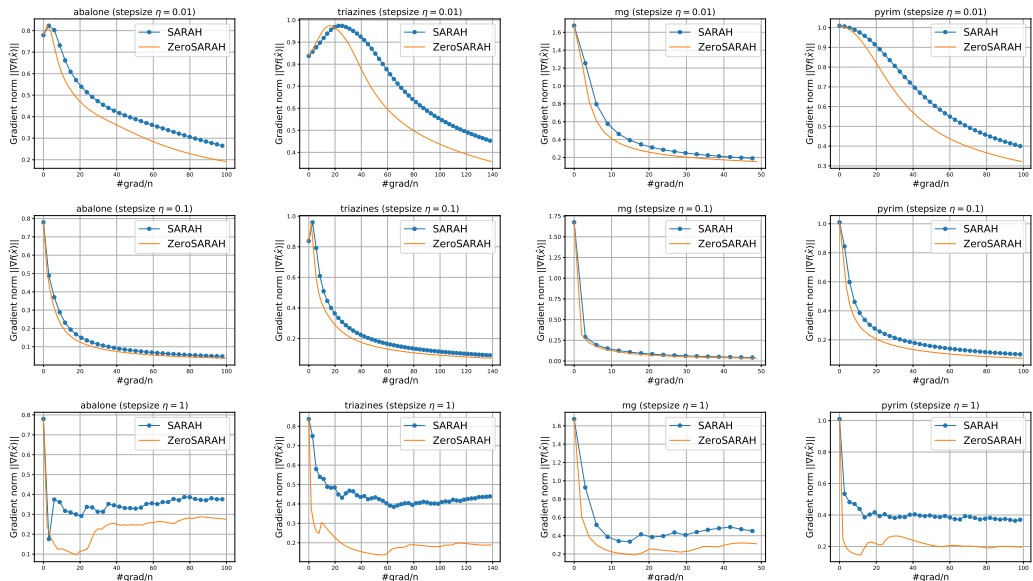

Figure 1: Performance between SARAH and ZeroSARAH under different datasets (columns) with respect to different stepsizes (rows). In rows, we have stepsizes $0.01, 0.1, 1$, respectively. In columns, we have LIBSVM datasates 'abalone', 'triazines', 'mg', 'pyrim', respectively.

In Figure 1, the $x$-axis and $y$-axis represent the number of stochastic gradient computations and the norm of gradient, respectively. The numerical results presented in Figure 1 are conducted on different datasets with different stepsizes. Regrading the parameter settings, we directly use the theoretical values according to the theorems or corollaries of SARAH and ZeroSARAH, i.e., we do not tune the parameters. Concretely, for SARAH (Algorithm 1), the epoch length $l = \sqrt{n}$ and the minibatch size $b = \sqrt{n}$ (see Theorem 6 in Pham et al. (2019)). For ZeroSARAH (Algorithm 2), the minibatch size $b_k \equiv \sqrt{n}$ for any $k \geq 0$, $\lambda_0 = 1$ and $\lambda_k = \frac{b_k}{2n} \equiv \frac{1}{2\sqrt{n}}$ for any $k \geq 1$ (see our Corollary 2). Note that there is no epoch length $l$ for ZeroSARAH since it is a loopless (single-loop) algorithm while SARAH requires $l$ for setting the length of its inner-loop (see Line 6 of Algorithm 1). For the stepsize $\eta$, both SARAH and ZeroSARAH adopt the same constant stepsize $\eta = O(\frac{1}{L})$. However the smooth parameter $L$ is not known in the experiments, thus here we use three stepsizes, i.e., $\eta = 0.01, 0.1, 1$.

**Remark:** The experimental results validate our theoretical convergence results (our ZeroSARAH can be slightly better than SARAH (see Table 1)) and confirm the practical superiority of ZeroSARAH (*avoid any full gradient computations*). To demonstrate the full gradient computations in Figure 1, we point out that *each circle marker in the curve of SARAH (blue curves) denotes a full gradient computation in SARAH. We emphasize that our* ZeroSARAH *never computes any full gradients*. Note that in this section we only present the experiments for the standard/centralized setting (1). Similar experiments in the distributed setting (2) are provided in Appendix A.2, e.g., Figure 2 demonstrates similar performance between distributed SARAH and distributed ZeroSARAH.

## 7   CONCLUSION

In this paper, we propose ZeroSARAH and its distributed variant D-ZeroSARAH algorithms for solving both standard and distributed nonconvex finite-sum problems (1) and (2). In particular, they are the first variance-reduced algorithms which do not require any full gradient computations, not even for the initial point. Moreover, our new algorithms can achieve better theoretical results than previous state-of-the-art results in certain regimes. While the numerical performance of our algorithms is also comparable/better than previous state-of-the-art algorithms, the main advantage of our algorithms is that they do not need to compute any full gradients. This characteristic can lead to practical significance of our algorithms since periodic computation of full gradient over all data samples from all clients usually is impractical and unaffordable.

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

# A  EXTRA EXPERIMENTS

In this appendix, we first describe the details of the objective functions and datasets used in our experiments in Appendix A.1. Then in Appendix A.2, we present the experimental results in the distributed setting (2).

## A.1  OBJECTIVE FUNCTIONS AND DATASETS IN EXPERIMENTS

The nonconvex robust linear regression problem (used in Wang et al. (2018)) is:

$$\min_{x \in \mathbb{R}^d} \left\{ f(x) := \frac{1}{n} \sum_{i=1}^{n} \ell(b_i - x^T a_i) \right\}, \tag{7}$$

where the nonconvex loss function $\ell(x) := \log(\frac{x^2}{2} + 1)$.

The binary classification with two-layer neural networks (used in Zhao et al. (2010); Tran-Dinh et al. (2019)) is:

$$\min_{x \in \mathbb{R}^d} \left\{ f(x) := \frac{1}{n} \sum_{i=1}^{n} \ell\left(a_i^T x, b_i\right) + \frac{\lambda}{2} \|x\|^2 \right\} \tag{8}$$

where $\{a_i\} \in \mathbb{R}^d, b_i \in \{-1, 1\}$, $\lambda \geq 0$ is an $\ell_2$-regularization parameter, and the function $\ell$ is defined as

$$\ell(x, y) := \left(1 - \frac{1}{1 + \exp(-xy)}\right)^2.$$

All datasets are downloaded from LIBSVM (Chang & Lin, 2011). The summary of datasets information is provided in the following Table 3.

Table 3: Metadata of datasets

| Dataset | $n$ (# of datapoints) | $d$ (# of features) |
|---------|----------------------|---------------------|
| a9a | 32561 | 123 |
| abalone | 4177 | 8 |
| mg | 1385 | 6 |
| mushrooms | 8124 | 112 |
| phishing | 11055 | 68 |
| pyrim | 74 | 27 |
| triazines | 186 | 60 |
| w8a | 49749 | 300 |

## A.2  EXPERIMENTS FOR THE DISTRIBUTED SETTING

Before presenting the experimental results in the distributed setting (2), we also need a distributed variant of SARAH-type methods in order to compare with our distributed variant of ZeroSARAH. Here we describe one possible version in Algorithm 4. Note that distributed SARAH also requires to periodically computes full gradients (see Line 7 of Algorithm 4), but it is not required by our D-ZeroSARAH (Algorithm 3).

In order to mimic distributed setup, we represented clients as parallel processes. We implement the training process using Python 3.8.8, mpi4py library. We run it on the workstation with 48 Cores, Intel(R) Xeon(R) Gold 6246 CPU @ 3.30GHz. We partition the dataset among 10 threads; having $M$ datapoints and $n$ clients, $k$-thread gets datapoints in range $k\lfloor \frac{M}{n} \rfloor + 1, \ldots, (k+1)\lfloor \frac{M}{n} \rfloor$. In case of $n\lfloor \frac{M}{n} \rfloor + 1 \leq M$, datapoints $n\lfloor \frac{M}{n} \rfloor + 1, \ldots M$ are ignored.

---

**Algorithm 4** Distributed SARAH-type methods (one possible version)

---

**Input:** initial point $x^0$, epoch length $l$, stepsize $\eta$, client minibatch size $s$, data minibatch size $b$
1: $x^{-1} = x^0$
2: **for** $k = 0, 1, 2, \ldots$ **do**
3:    **if** $k \mod l = 0$ **then**
4:       **for** each client $i \in \{1, \ldots, n\}$ **do**
5:          Compute full gradient of each client: $g_i^k = \frac{1}{m} \sum_{j=1}^m \nabla f_{i,j}(x^k)$        $// = \nabla f_i(x^k)$
6:       **end for**
7:       $v^k = \frac{1}{n} \sum_{i=1}^n g_i^k$        // full gradient computations
8:    **else**
9:       Randomly sample a subset of clients $S^k$ from $n$ clients with size $|S^k| = s$
10:      **for** each client $i \in S^k$ **do**
11:      Sample minibatch $I_i^k$ of size $|I_i^k| = b$ (from the $m$ data samples in client $i$)
12:      Compute the local minibatch gradient information:
          $g_{i,\mathrm{curr}}^k = \frac{1}{b} \sum_{j \in I_i^k} \nabla f_{i,j}(x^k)$   and   $g_{i,\mathrm{prev}}^k = \frac{1}{b} \sum_{j \in I_i^k} \nabla f_{i,j}(x^{k-1})$
13:      **end for**
14:      $v^k = \frac{1}{s} \sum_{i \in S^k} \left( g_{i,\mathrm{curr}}^k - g_{i,\mathrm{prev}}^k \right) + v^{k-1}$
15:    **end if**
16:    $x^{k+1} = x^k - \eta_k v^k$
17: **end for**

---

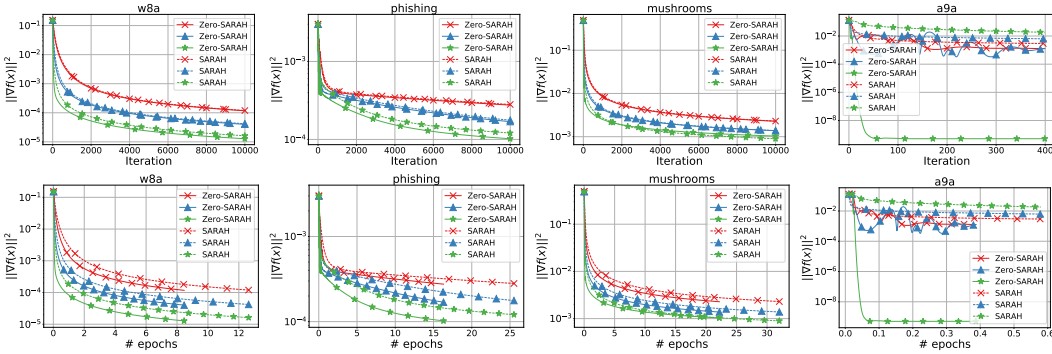

Figure 2: Performance between distributed SARAH and distributed ZeroSARAH under different datasets (columns). We show convergence using the theoretical stepsize in red lines; in blue lines, we scale it by factor $\times 3$; in green lines, we scale it by factor $\times 9$.

In Figure 2, we present numerical results for the distributed setting. The solid lines and dashed lines denote the distributed ZeroSARAH and distributed SARAH, respectively. Regrading the parameter settings, we also use the theoretical values according to the theorems or corollaries. In particular, from Tran-Dinh et al. (2019), we know that the smoothness constant $L \approx 0.15405 \max_i \|a_i\|^2 + \lambda$ for objective function (8). We choose the regularizer parameter to be $\lambda = 0.15405 \cdot 10^{-6} \max_i \|a_i\|^2$. In order to obtain comparable plots similar to the standard/centralized setting (Figure 1), we also use multiple stepsizes. We choose the theoretical stepsize from Corollary 4 (i.e., $\frac{1}{(1+\sqrt{8})L}$) scaled by factors of $\times 1$ (red curves), $\times 3$ (blue curves), $\times 9$ (green curves), respectively.

**Remark:** Similar to Figure 1, the experimental results in Figure 2 also validate our theoretical convergence results (distributed ZeroSARAH (D-ZeroSARAH) can be slightly better than distributed SARAH (see Table 2)) and confirm the practical superiority of D-ZeroSARAH (avoid any full gradient computations) for the distributed setting (2).

# B  MISSING PROOFS FOR ZeroSARAH

In this appendix, we provide the missing proofs for the standard nonconvex setting (1). Concretely, we provide the detailed proofs for Theorem 1 and Corollaries 1–2 of ZeroSARAH in Section 4.

## B.1  PROOF OF THEOREM 1

First, we need a useful lemma in Li et al. (2021) which describes the relation between the function values after and before a gradient descent step.

**Lemma 1 (Li et al. (2021))** *Suppose that function $f$ is $L$-smooth and let $x^{k+1} := x^k - \eta_k v^k$. Then for any $v^k \in \mathbb{R}^d$ and $\eta_k > 0$, we have*

$$f(x^{k+1}) \leq f(x^k) - \frac{\eta_k}{2}\|\nabla f(x^k)\|^2 - \left(\frac{1}{2\eta_k} - \frac{L}{2}\right)\|x^{k+1} - x^k\|^2 + \frac{\eta_k}{2}\|v^k - \nabla f(x^k)\|^2. \quad (9)$$

Then, we provide the following Lemma 2 to bound the last variance term of (9).

**Lemma 2** *Suppose that Assumption 1 holds. The gradient estimator $v^k$ is defined in Line 5 of Algorithm 2, then we have*

$$\mathbb{E}_k[\|v^k - \nabla f(x^k)\|^2] \leq (1 - \lambda_k)^2\|v^{k-1} - \nabla f(x^{k-1})\|^2 + \frac{2L^2}{b_k}\|x^k - x^{k-1}\|^2$$

$$+ \frac{2\lambda_k^2}{b_k}\frac{1}{n}\sum_{j=1}^n \|\nabla f_j(x^{k-1}) - y_j^{k-1}\|^2. \quad (10)$$

***Proof of Lemma 2.*** First, according to the gradient estimator $v^k$ of ZeroSARAH (see Line 5 of Algorithm 2), we know that

$$v^k = \frac{1}{b_k}\sum_{i\in I_b^k}\left(\nabla f_i(x^k) - \nabla f_i(x^{k-1})\right) + (1-\lambda_k)v^{k-1} + \lambda_k\left(\frac{1}{b_k}\sum_{i\in I_b^k}\left(\nabla f_i(x^{k-1}) - y_i^{k-1}\right) + \frac{1}{n}\sum_{j=1}^n y_j^{k-1}\right) \quad (11)$$

Now we bound the variance as follows:

$$\mathbb{E}_k[\|v^k - \nabla f(x^k)\|^2]$$

$$\stackrel{(11)}{=} \mathbb{E}_k\left[\left\|\frac{1}{b_k}\sum_{i\in I_b^k}\left(\nabla f_i(x^k) - \nabla f_i(x^{k-1})\right) + (1-\lambda_k)v^{k-1}\right.\right.$$

$$\left.\left. + \lambda_k\left(\frac{1}{b_k}\sum_{i\in I_b^k}\left(\nabla f_i(x^{k-1}) - y_i^{k-1}\right) + \frac{1}{n}\sum_{j=1}^n y_j^{k-1}\right) - \nabla f(x^k)\right\|^2\right]$$

$$= \mathbb{E}_k\left[\left\|\frac{1}{b_k}\sum_{i\in I_b^k}\left(\nabla f_i(x^k) - \nabla f_i(x^{k-1})\right) + \nabla f(x^{k-1}) - \nabla f(x^k) + (1-\lambda_k)(v^{k-1} - \nabla f(x^{k-1}))\right.\right.$$

$$\left.\left. + \lambda_k\left(\frac{1}{b_k}\sum_{i\in I_b^k}\left(\nabla f_i(x^{k-1}) - y_i^{k-1}\right) + \frac{1}{n}\sum_{j=1}^n y_j^{k-1} - \nabla f(x^{k-1})\right)\right\|^2\right]$$

$$= \mathbb{E}_k\left[\left\|\frac{1}{b_k}\sum_{i\in I_b^k}\left(\nabla f_i(x^k) - \nabla f_i(x^{k-1})\right) + \nabla f(x^{k-1}) - \nabla f(x^k)\right.\right.$$

$$\left.\left. + \lambda_k\left(\frac{1}{b_k}\sum_{i\in I_b^k}\left(\nabla f_i(x^{k-1}) - y_i^{k-1}\right) + \frac{1}{n}\sum_{j=1}^n y_j^{k-1} - \nabla f(x^{k-1})\right)\right\|^2\right]$$

$$+ (1-\lambda_k)^2\|v^{k-1} - \nabla f(x^{k-1})\|^2$$

$$\leq 2\mathbb{E}_k\left[\left\|\frac{1}{b_k}\sum_{i\in I_b^k}\left(\nabla f_i(x^k)-\nabla f_i(x^{k-1})\right)+\nabla f(x^{k-1})-\nabla f(x^k)\right\|^2\right]$$

$$+2\mathbb{E}_k\left[\lambda_k^2\left\|\frac{1}{b_k}\sum_{i\in I_b^k}\left(\nabla f_i(x^{k-1})-y_i^{k-1}\right)+\frac{1}{n}\sum_{j=1}^n y_j^{k-1}-\nabla f(x^{k-1})\right\|^2\right]$$

$$+(1-\lambda_k)^2\|v^{k-1}-\nabla f(x^{k-1})\|^2$$

$$\leq\frac{2L^2}{b_k}\|x^k-x^{k-1}\|^2+\frac{2\lambda_k^2}{b_k}\frac{1}{n}\sum_{j=1}^n\|\nabla f_j(x^{k-1})-y_j^{k-1}\|^2+(1-\lambda_k)^2\|v^{k-1}-\nabla f(x^{k-1})\|^2,$$

$$(12)$$

where (12) uses the $L$-smoothness Assumption 1 and the fact that $\mathbb{E}[\|x-\mathbb{E}x\|^2]\leq\mathbb{E}[\|x\|^2]$, for any random variable $x$. $\qquad\square$

To deal with the last term of (10), we use the following Lemma 3.

**Lemma 3** *Suppose that Assumption 1 holds. The update of $\{y_i^k\}$ is defined in Line 7 of Algorithm 2, then we have, for $\forall\beta_k>0$,*

$$\mathbb{E}_k\left[\frac{1}{n}\sum_{j=1}^n\|\nabla f_j(x^k)-y_j^k\|^2\right]\leq\left(1-\frac{b_k}{n}\right)(1+\beta_k)\frac{1}{n}\sum_{j=1}^n\|\nabla f_j(x^{k-1})-y_j^{k-1}\|^2$$

$$+\left(1-\frac{b_k}{n}\right)\left(1+\frac{1}{\beta_k}\right)L^2\|x^k-x^{k-1}\|^2.\qquad(13)$$

***Proof of Lemma 3.*** According to the update of $\{y_i^k\}$ (see Line 7 of Algorithm 2), we have

$$\mathbb{E}_k\left[\frac{1}{n}\sum_{j=1}^n\|\nabla f_j(x^k)-y_j^k\|^2\right]$$

$$=\left(1-\frac{b_k}{n}\right)\frac{1}{n}\sum_{j=1}^n\|\nabla f_j(x^k)-y_j^{k-1}\|^2\qquad(14)$$

$$=\left(1-\frac{b_k}{n}\right)\frac{1}{n}\sum_{j=1}^n\|\nabla f_j(x^k)-\nabla f_j(x^{k-1})+\nabla f_j(x^{k-1})-y_j^{k-1}\|^2$$

$$\leq\left(1-\frac{b_k}{n}\right)(1+\beta_k)\frac{1}{n}\sum_{j=1}^n\|\nabla f_j(x^{k-1})-y_j^{k-1}\|^2+\left(1-\frac{b_k}{n}\right)\left(1+\frac{1}{\beta_k}\right)L^2\|x^k-x^{k-1}\|^2,\quad(15)$$

where (14) uses the update of $\{y_j^k\}$ (see Line 7 of Algorithm 2), and (15) uses Young's inequality and $L$-smoothness Assumption 1. $\qquad\square$

Now we combine Lemmas 1–3 (i.e., (9), (10) and (13)) to prove Theorem 1.

***Proof of Theorem 1.*** First, we take expectation to obtain

$$\mathbb{E}\left[f(x^{k+1})-f^*+\left(\gamma_k-\frac{\eta_k}{2}\right)\|v^k-\nabla f(x^k)\|^2+\left(\frac{1}{2\eta_k}-\frac{L}{2}\right)\|x^{k+1}-x^k\|^2+\alpha_k\frac{1}{n}\sum_{j=1}^n\|\nabla f_j(x^k)-y_j^k\|^2\right]$$

$$\leq\mathbb{E}\left[f(x^k)-f^*-\frac{\eta_k}{2}\|\nabla f(x^k)\|^2+\gamma_k(1-\lambda_k)^2\|v^{k-1}-\nabla f(x^{k-1})\|^2\right.$$

$$+\frac{2\gamma_k L^2}{b_k}\|x^k-x^{k-1}\|^2+\frac{2\gamma_k\lambda_k^2}{b_k}\frac{1}{n}\sum_{j=1}^n\|\nabla f_j(x^{k-1})-y_j^{k-1}\|^2$$

$$\left.+\alpha_k\left(1-\frac{b_k}{n}\right)\left(1+\frac{1}{\beta_k}\right)L^2\|x^k-x^{k-1}\|^2+\alpha_k\left(1-\frac{b_k}{n}\right)(1+\beta_k)\frac{1}{n}\sum_{j=1}^n\|\nabla f(x^{k-1})-y_j^{k-1}\|^2\right]$$

$$= \mathbb{E}\left[f(x^k) - f^* - \frac{\eta_k}{2}\|\nabla f(x^k)\|^2 + \gamma_k(1-\lambda_k)^2\|v^{k-1} - \nabla f(x^{k-1})\|^2 \right.$$

$$+ \left(\frac{2\gamma_k L^2}{b_k} + \alpha_k(1 - \frac{b_k}{n})(1 + \frac{1}{\beta_k})L^2\right)\|x^k - x^{k-1}\|^2$$

$$\left. + \left(\frac{2\gamma_k \lambda_k^2}{b_k} + \alpha_k(1 - \frac{b_k}{n})(1 + \beta_k)\right)\frac{1}{n}\sum_{j=1}^n \|\nabla f_j(x^{k-1}) - y_j^{k-1}\|^2\right]. \tag{16}$$

Now we choose appropriate parameters. Let $\gamma_k = \frac{\eta_{k-1}}{2\lambda_k}$ and $\gamma_k \leq \gamma_{k-1}$, then $\gamma_k(1-\lambda_k)^2 \leq \gamma_{k-1} - \frac{\eta_{k-1}}{2}$. Let $\beta_k = \frac{b_k}{2n}$, $\alpha_k = \frac{2n\lambda_k\eta_{k-1}}{b_k^2}$ and $\alpha_k \leq \alpha_{k-1}$, we have $\frac{2\gamma_k\lambda_k^2}{b_k} + \alpha_k(1-\frac{b_k}{n})(1+\beta_k) \leq \alpha_{k-1}$. We also have $\frac{2\gamma_k L^2}{b_k} + \alpha_k(1-\frac{b_k}{n})(1+\frac{1}{\beta_k})L^2 \leq \frac{1}{2\eta_{k-1}} - \frac{L}{2}$ by further letting stepsize

$$\eta_{k-1} \leq \frac{1}{L(1 + \sqrt{M_k})}, \tag{17}$$

where $M_k := \frac{2}{\lambda_k b_k} + \frac{8\lambda_k n^2}{b_k^3}$.

Summing up (16) from $k = 1$ to $K - 1$, we get

$$0 \leq \mathbb{E}\left[f(x^1) - f^* - \sum_{k=1}^{K-1}\frac{\eta_k}{2}\|\nabla f(x^k)\|^2 + \gamma_1(1-\lambda_1)^2\|v^0 - \nabla f(x^0)\|^2 \right.$$

$$+ \left(\frac{2\gamma_1 L^2}{b_1} + \alpha_1(1-\frac{b_1}{n})(1+\frac{2n}{b_1})L^2\right)\|x^1 - x^0\|^2$$

$$\left. + \left(\frac{2\gamma_1\lambda_1^2}{b_1} + \alpha_1(1-\frac{b_1}{n})(1+\frac{b_1}{2n})\right)\frac{1}{n}\sum_{j=1}^n\|\nabla f_j(x^0) - y_j^0\|^2\right]. \tag{18}$$

For $k = 0$, we directly uses (9), i.e.,

$$\mathbb{E}[f(x^1) - f^*] \leq \mathbb{E}\left[f(x^0) - f^* - \frac{\eta_0}{2}\|\nabla f(x^0)\|^2 - \left(\frac{1}{2\eta_0} - \frac{L}{2}\right)\|x^1 - x^0\|^2 + \frac{\eta_0}{2}\|v^0 - \nabla f(x^0)\|^2\right]. \tag{19}$$

Now, we combine (18) and (19) to get

$$\mathbb{E}\left[\sum_{k=0}^{K-1}\frac{\eta_k}{2}\|\nabla f(x^k)\|^2\right]$$

$$\leq \mathbb{E}\left[f(x^0) - f^* + \left(\gamma_1(1-\lambda_1)^2 + \frac{\eta_0}{2}\right)\|v^0 - \nabla f(x^0)\|^2 \right.$$

$$\left. + \left(\frac{2\gamma_1\lambda_1^2}{b_1} + \alpha_1(1-\frac{b_1}{n})(1+\frac{b_1}{2n})\right)\frac{1}{n}\sum_{j=1}^n\|\nabla f_j(x^0) - y_j^0\|^2\right] \tag{20}$$

$$\leq \mathbb{E}\left[f(x^0) - f^* + \frac{\eta_0(1-\lambda_1(1-\lambda_1))}{2\lambda_1}\|v^0 - \nabla f(x^0)\|^2 + \frac{2n\lambda_1\eta_0}{b_1^2}\frac{1}{n}\sum_{j=1}^n\|\nabla f_j(x^0) - y_j^0\|^2\right] \tag{21}$$

$$\leq \mathbb{E}\left[f(x^0) - f^* + \gamma_0\|v^0 - \nabla f(x^0)\|^2 + \alpha_0\frac{1}{n}\sum_{j=1}^n\|\nabla f_j(x^0) - y_j^0\|^2\right] \tag{22}$$

$$\leq f(x^0) - f^* + \gamma_0\frac{n-b_0}{(n-1)b_0}\frac{1}{n}\sum_{j=1}^n\|\nabla f_j(x^0)\|^2 + \alpha_0(1-\frac{b_0}{n})\frac{1}{n}\sum_{j=1}^n\|\nabla f_j(x^0)\|^2 \tag{23}$$

$$= f(x^0) - f^* + \left(\gamma_0\frac{n-b_0}{(n-1)b_0} + \alpha_0(1-\frac{b_0}{n})\right)\frac{1}{n}\sum_{j=1}^n\|\nabla f_j(x^0)\|^2, \tag{24}$$

where (20) follows from the definition of $\eta_0$ in (17), (21) uses $\gamma_1 = \frac{\eta_0}{2\lambda_1}$ and $\alpha_1 = \frac{2n\lambda_1\eta_0}{b_1^2}$, (22) holds by choosing $\gamma_0 \geq \frac{\eta_0}{2\lambda_1} \geq \frac{\eta_0(1-\lambda_1(1-\lambda_1))}{2\lambda_1}$ and $\alpha_0 \geq \frac{2n\lambda_1\eta_0}{b_1^2}$, and (23) uses $\lambda_0 = 1$. By randomly choosing $\widehat{x}^K$ from $\{x^k\}_{k=0}^{K-1}$ with probability $\eta_k / \sum_{t=0}^{K-1} \eta_t$ for $x^k$, (24) turns to

$$\mathbb{E}[\|\nabla f(\widehat{x}^K)\|^2] \leq \frac{2(f(x^0) - f^*)}{\sum_{k=0}^{K-1} \eta_k} + \frac{2}{\sum_{k=0}^{K-1} \eta_k}\Big(\gamma_0 \frac{n - b_0}{(n-1)b_0} + \alpha_0(1 - \frac{b_0}{n})\Big)\frac{1}{n}\sum_{j=1}^{n} \|\nabla f_j(x^0)\|^2$$

$$\leq \frac{2(f(x^0) - f^*)}{\sum_{k=0}^{K-1} \eta_k} + \frac{(n - b_0)(4\gamma_0 + 2\alpha_0 b_0)}{nb_0 \sum_{k=0}^{K-1} \eta_k}\frac{1}{n}\sum_{j=1}^{n} \|\nabla f_j(x^0)\|^2. \tag{25}$$

$\square$

### B.2 PROOFS OF COROLLARIES 1 AND 2

Now, we prove the detailed convergence results in Corollaries 1–2 with specific parameter settings.

***Proof of Corollary 1.*** First we know that (25) with $b_0 = n$ turns to

$$\mathbb{E}[\|\nabla f(\widehat{x}^K)\|^2] \leq \frac{2(f(x^0) - f^*)}{\sum_{k=0}^{K-1} \eta_k}. \tag{26}$$

Then if we set $\lambda_k = \frac{b_k}{2n}$ and $b_k \equiv \sqrt{n}$ for any $k \geq 1$, then we know that $M_k := \frac{2}{\lambda_k b_k} + \frac{8\lambda_k n^2}{b_k^3} \equiv 8$ and thus the stepsize $\eta_k \leq \frac{1}{L\left(1 + \sqrt{M_{k+1}}\right)} \equiv \frac{1}{(1+\sqrt{8})L}$ for any $k \geq 0$. By plugging $\eta_k \leq \frac{1}{(1+\sqrt{8})L}$ into (26), we have

$$\mathbb{E}[\|\nabla f(\widehat{x}^K)\|^2] \leq \frac{2(1 + \sqrt{8})L(f(x^0) - f^*)}{K} = \epsilon^2,$$

where the last equality holds by letting the number of iterations $K = \frac{2(1+\sqrt{8})L(f(x^0)-f^*)}{\epsilon^2}$. Thus the number of stochastic gradient computations is

$$\#\text{grad} = \sum_{k=0}^{K-1} b_k = b_0 + \sum_{k=1}^{K-1} b_k = n + (K-1)\sqrt{n} \leq n + \frac{2(1 + \sqrt{8})\sqrt{n}L(f(x^0) - f^*)}{\epsilon^2}.$$

$\square$

***Proof of Corollary 2.*** First we recall (25) here:

$$\mathbb{E}[\|\nabla f(\widehat{x}^K)\|^2] \leq \frac{2(f(x^0) - f^*)}{\sum_{k=0}^{K-1} \eta_k} + \frac{(n - b_0)(4\gamma_0 + 2\alpha_0 b_0)}{nb_0 \sum_{k=0}^{K-1} \eta_k}\frac{1}{n}\sum_{j=1}^{n} \|\nabla f_j(x^0)\|^2. \tag{27}$$

In this corollary, we do not compute any full gradients even for the initial point. We set the minibatch size $b_k \equiv \sqrt{n}$ for any $k \geq 0$. So we need consider the second term of (27) since $b_0 = \sqrt{n}$ is not equal to $n$. Similar to Corollary 1, if we set $\lambda_k = \frac{b_k}{2n}$ for any $k \geq 1$, then we know that $M_k := \frac{2}{\lambda_k b_k} + \frac{8\lambda_k n^2}{b_k^3} \equiv 8$ and thus the stepsize $\eta_k \leq \frac{1}{L\left(1 + \sqrt{M_{k+1}}\right)} \equiv \frac{1}{(1+\sqrt{8})L}$ for any $k \geq 0$. For the second term, we recall that $\gamma_0 \geq \frac{\eta_0}{2\lambda_1} = \frac{\sqrt{n}}{(1+\sqrt{8})L}$ and $\alpha_0 \geq \frac{2n\lambda_1\eta_0}{b_1^2} = \frac{1}{(1+\sqrt{8})L\sqrt{n}}$. It is easy to see that $\gamma_0 \geq \alpha_0 b_0$ since $b_0 = \sqrt{n} \leq n$. Now, we can change (27) to

$$\mathbb{E}[\|\nabla f(\widehat{x}^K)\|^2] \leq \frac{2(1 + \sqrt{8})L(f(x^0) - f^*)}{K} + \frac{6(n - \sqrt{n})}{nK}\frac{1}{n}\sum_{j=1}^{n} \|\nabla f_j(x^0)\|^2$$

$$\leq \frac{2(1 + \sqrt{8})L(f(x^0) - f^*) + 6G_0}{K} \tag{28}$$

$$= \epsilon^2,$$

where (28) is due to the definition $G_0 := \frac{1}{n} \sum_{i=1}^{n} \|\nabla f_i(x^0)\|^2$, and the last equality holds by letting the number of iterations $K = \frac{2(1+\sqrt{8})L(f(x^0)-f^*)+6G_0}{\epsilon^2}$. Thus the number of stochastic gradient computations is

$$\#\text{grad} = \sum_{k=0}^{K-1} b_k = \sqrt{n}K = \sqrt{n}\frac{2(1+\sqrt{8})L(f(x^0)-f^*)+6G_0}{\epsilon^2} = O\left(\frac{\sqrt{n}(L\Delta_0 + G_0)}{\epsilon^2}\right).$$

Note that $G_0$ can be bounded by $G_0 \leq 2L(f(x^0) - \widehat{f}^*)$ via $L$-smoothness Assumption 1, then we have

$$\#\text{grad} = O\left(\frac{\sqrt{n}(L\Delta_0 + L\widehat{\Delta}_0)}{\epsilon^2}\right).$$

Note that $\Delta_0 := f(x^0) - f^*$, where $f^* := \min_x f(x)$, and $\widehat{\Delta}_0 := f(x^0) - \widehat{f}^*$, where $\widehat{f}^* := \frac{1}{n} \sum_{i=1}^{n} \min_x f_i(x)$.  □

## C  MISSING PROOFS FOR D-ZeroSARAH

In this appendix, we provide the missing proofs for the distributed nonconvex setting (2). Concretely, we provide the detailed proofs for Theorem 2 and Corollaries 3–4 of D-ZeroSARAH in Section 5.

### C.1  PROOF OF THEOREM 2

Similar to Appendix B.1, we first recall the lemma in Li et al. (2021) which describes the change of function value after a gradient update step.

**Lemma 1 (Li et al. (2021))** *Suppose that function $f$ is $L$-smooth and let $x^{k+1} := x^k - \eta_k v^k$. Then for any $v^k \in \mathbb{R}^d$ and $\eta_k > 0$, we have*

$$f(x^{k+1}) \leq f(x^k) - \frac{\eta_k}{2}\|\nabla f(x^k)\|^2 - \left(\frac{1}{2\eta_k} - \frac{L}{2}\right)\|x^{k+1} - x^k\|^2 + \frac{\eta_k}{2}\|v^k - \nabla f(x^k)\|^2. \quad (29)$$

Then, we provide the following Lemma 4 to bound the last variance term of (29).

**Lemma 4** *Suppose that Assumption 2 holds. The gradient estimator $v^k$ is defined in Line 9 of Algorithm 3, then we have*

$$\mathbb{E}_k[\|v^k - \nabla f(x^k)\|^2] \leq (1 - \lambda_k)^2 \|v^{k-1} - \nabla f(x^{k-1})\|^2 + \frac{2L^2}{s_k b_k}\|x^k - x^{k-1}\|^2$$

$$+ \frac{2\lambda_k^2}{s_k b_k} \frac{1}{nm} \sum_{i,j=1,1}^{n,m} \|\nabla f_{i,j}(x^{k-1}) - y_{i,j}^{k-1}\|^2. \quad (30)$$

***Proof of Lemma 4.*** First, according to the gradient estimator $v^k$ of D-ZeroSARAH (see Line 9 of Algorithm 3), we know that

$$v^k = \frac{1}{s_k} \sum_{i \in S^k} \left(g_{i,\text{curr}}^k - g_{i,\text{prev}}^k\right) + (1 - \lambda_k)v^{k-1} + \lambda_k \frac{1}{s_k} \sum_{i \in S^k} \left(g_{i,\text{prev}}^k - y_{i,\text{prev}}^k\right) + \lambda_k y^{k-1} \quad (31)$$

Now we bound the variance as follows:

$$\mathbb{E}_k[\|v^k - \nabla f(x^k)\|^2]$$

$$\overset{(31)}{=} \mathbb{E}_k\left[\left\|\frac{1}{s_k} \sum_{i \in S^k} \left(g_{i,\text{curr}}^k - g_{i,\text{prev}}^k\right) + (1 - \lambda_k)v^{k-1} + \lambda_k\left(\frac{1}{s_k} \sum_{i \in S^k} \left(g_{i,\text{prev}}^k - y_{i,\text{prev}}^k\right) + y^{k-1}\right) - \nabla f(x^k)\right\|^2\right]$$

$$= \mathbb{E}_k\left[\left\|\frac{1}{s_k} \sum_{i \in S^k} \left(g_{i,\text{curr}}^k - g_{i,\text{prev}}^k\right) + \nabla f(x^{k-1}) - \nabla f(x^k) + (1 - \lambda_k)(v^{k-1} - \nabla f(x^{k-1}))\right.\right.$$

$$+ \lambda_k \Big( \frac{1}{s_k} \sum_{i \in S^k} \big( g^k_{i,\text{prev}} - y^k_{i,\text{prev}} \big) + y^{k-1} - \nabla f(x^{k-1}) \Big) \Big\|^2 \Big]$$

$$= \mathbb{E}_k \Big[ \Big\| \frac{1}{s_k} \sum_{i \in S^k} \big( g^k_{i,\text{curr}} - g^k_{i,\text{prev}} \big) + \nabla f(x^{k-1}) - \nabla f(x^k)$$

$$+ \lambda_k \Big( \frac{1}{s_k} \sum_{i \in S^k} \big( g^k_{i,\text{prev}} - y^k_{i,\text{prev}} \big) + y^{k-1} - \nabla f(x^{k-1}) \Big) \Big\|^2 \Big]$$

$$+ (1 - \lambda_k)^2 \| v^{k-1} - \nabla f(x^{k-1}) \|^2$$

$$\leq 2 \mathbb{E}_k \Big[ \Big\| \frac{1}{s_k b_k} \sum_{i \in S^k} \sum_{j \in I^k_{b_i}} \big( \nabla f_{i,j}(x^k) - \nabla f_{i,j}(x^{k-1}) \big) + \nabla f(x^{k-1}) - \nabla f(x^k) \Big\|^2 \Big]$$

$$+ 2 \mathbb{E}_k \Big[ \lambda_k^2 \Big\| \frac{1}{s_k b_k} \sum_{i \in S^k} \sum_{j \in I^k_{b_i}} \big( \nabla f_{i,j}(x^{k-1}) - y^{k-1}_{i,j} \big) + y^{k-1} - \nabla f(x^{k-1}) \Big\|^2 \Big]$$

$$+ (1 - \lambda_k)^2 \| v^{k-1} - \nabla f(x^{k-1}) \|^2$$

$$\leq \frac{2L^2}{s_k b_k} \| x^k - x^{k-1} \|^2 + \frac{2\lambda_k^2}{s_k b_k} \frac{1}{nm} \sum_{i,j=1,1}^{n,m} \| \nabla f_{i,j}(x^{k-1}) - y^{k-1}_{i,j} \|^2 + (1 - \lambda_k)^2 \| v^{k-1} - \nabla f(x^{k-1}) \|^2,$$

$$(32)$$

where (32) uses the $L$-smoothness Assumption 2, i.e., $\| \nabla f_{i,j}(x) - \nabla f_{i,j}(y) \| \leq L \| x - y \|$, and the fact that $\mathbb{E}[\| x - \mathbb{E}x \|^2] \leq \mathbb{E}[\| x \|^2]$ for any random variable $x$. □

To deal with the last term in (30), we uses the following Lemma 5.

**Lemma 5** *Suppose that Assumption 2 holds. The update of $\{y^k_{i,j}\}$ is defined in Line 7 of Algorithm 3, then we have, for $\forall \beta_k > 0$,*

$$\mathbb{E}_k \Big[ \frac{1}{nm} \sum_{i,j=1,1}^{n,m} \| \nabla f_{i,j}(x^k) - y^k_{i,j} \|^2 \Big] \leq \big( 1 - \frac{s_k b_k}{nm} \big)(1 + \beta_k) \frac{1}{nm} \sum_{i,j=1,1}^{n,m} \| \nabla f_{i,j}(x^{k-1}) - y^{k-1}_{i,j} \|^2$$

$$+ \big( 1 - \frac{s_k b_k}{nm} \big)\big( 1 + \frac{1}{\beta_k} \big) L^2 \| x^k - x^{k-1} \|^2. \quad (33)$$

***Proof of Lemma 5.*** According to the update of $\{y^k_{i,j}\}$ (see Line 7 and Line 11 of Algorithm 3), we have

$$\mathbb{E}_k \Big[ \frac{1}{nm} \sum_{i,j=1,1}^{n,m} \| \nabla f_{i,j}(x^k) - y^k_{i,j} \|^2 \Big]$$

$$= \big( 1 - \frac{s_k b_k}{nm} \big) \frac{1}{nm} \sum_{i,j=1,1}^{n,m} \| \nabla f_{i,j}(x^k) - y^{k-1}_{i,j} \|^2 \quad (34)$$

$$= \big( 1 - \frac{s_k b_k}{nm} \big) \frac{1}{nm} \sum_{i,j=1,1}^{n,m} \| \nabla f_{i,j}(x^k) - \nabla f_{i,j}(x^{k-1}) + \nabla f_{i,j}(x^{k-1}) - y^{k-1}_{i,j} \|^2$$

$$\leq \big( 1 - \frac{s_k b_k}{nm} \big)(1 + \beta_k) \frac{1}{nm} \sum_{i,j=1,1}^{n,m} \| \nabla f_{i,j}(x^{k-1}) - y^{k-1}_{i,j} \|^2 + \big( 1 - \frac{s_k b_k}{nm} \big)\big( 1 + \frac{1}{\beta_k} \big) L^2 \| x^k - x^{k-1} \|^2,$$

$$(35)$$

where (34) uses the update of $\{y^k_{i,j}\}$ in Algorithm 3, and (35) uses Young's inequality and $L$-smoothness Assumption 2. □

Now we combine Lemmas 1, 4 and 5 (i.e., (29), (30) and (33)) to prove Theorem 2.

***Proof of Theorem 2.*** First, we take expectation to obtain

$$
\mathbb{E}\left[ f(x^{k+1}) - f^* + \left(\gamma_k - \frac{\eta_k}{2}\right)\|v^k - \nabla f(x^k)\|^2 + \left(\frac{1}{2\eta_k} - \frac{L}{2}\right)\|x^{k+1} - x^k\|^2 \right.
$$

$$
\left. + \alpha_k \frac{1}{nm} \sum_{i,j=1,1}^{n,m} \|\nabla f_{i,j}(x^k) - y_{i,j}^k\|^2 \right]
$$

$$
\leq \mathbb{E}\left[ f(x^k) - f^* - \frac{\eta_k}{2}\|\nabla f(x^k)\|^2 + \gamma_k(1-\lambda_k)^2\|v^{k-1} - \nabla f(x^{k-1})\|^2 \right.
$$

$$
+ \frac{2\gamma_k L^2}{s_k b_k}\|x^k - x^{k-1}\|^2 + \frac{2\gamma_k \lambda_k^2}{s_k b_k} \frac{1}{nm} \sum_{i,j=1,1}^{n,m} \|\nabla f_{i,j}(x^{k-1}) - y_{i,j}^{k-1}\|^2
$$

$$
+ \alpha_k\left(1 - \frac{s_k b_k}{nm}\right)\left(1 + \frac{1}{\beta_k}\right)L^2\|x^k - x^{k-1}\|^2
$$

$$
\left. + \alpha_k\left(1 - \frac{s_k b_k}{nm}\right)(1 + \beta_k)\frac{1}{nm} \sum_{i,j=1,1}^{n,m} \|\nabla f_{i,j}(x^{k-1}) - y_{i,j}^{k-1}\|^2 \right]
$$

$$
= \mathbb{E}\left[ f(x^k) - f^* - \frac{\eta_k}{2}\|\nabla f(x^k)\|^2 + \gamma_k(1-\lambda_k)^2\|v^{k-1} - \nabla f(x^{k-1})\|^2 \right.
$$

$$
+ \left(\frac{2\gamma_k L^2}{s_k b_k} + \alpha_k\left(1 - \frac{s_k b_k}{nm}\right)\left(1 + \frac{1}{\beta_k}\right)L^2\right)\|x^k - x^{k-1}\|^2
$$

$$
\left. + \left(\frac{2\gamma_k \lambda_k^2}{s_k b_k} + \alpha_k\left(1 - \frac{s_k b_k}{nm}\right)(1 + \beta_k)\right)\frac{1}{nm} \sum_{i,j=1,1}^{n,m} \|\nabla f_{i,j}(x^{k-1}) - y_{i,j}^{k-1}\|^2 \right]. \quad (36)
$$

Now we choose appropriate parameters. Let $\gamma_k = \frac{\eta_{k-1}}{2\lambda_k}$ and $\gamma_k \leq \gamma_{k-1}$, then $\gamma_k(1-\lambda_k)^2 \leq \gamma_{k-1} - \frac{\eta_{k-1}}{2}$. Let $\beta_k = \frac{s_k b_k}{2nm}$, $\alpha_k = \frac{2nm\lambda_k \eta_{k-1}}{s_k^2 b_k^2}$ and $\alpha_k \leq \alpha_{k-1}$, we have $\frac{2\gamma_k \lambda_k^2}{s_k b_k} + \alpha_k\left(1 - \frac{s_k b_k}{nm}\right)(1+\beta_k) \leq \alpha_{k-1}$. We also have $\frac{2\gamma_k L^2}{s_k b_k} + \alpha_k\left(1 - \frac{s_k b_k}{nm}\right)\left(1 + \frac{1}{\beta_k}\right)L^2 \leq \frac{1}{2\eta_{k-1}} - \frac{L}{2}$ by further letting stepsize

$$
\eta_{k-1} \leq \frac{1}{L\left(1 + \sqrt{W_k}\right)}, \quad (37)
$$

where $W_k := \frac{2}{\lambda_k s_k b_k} + \frac{8\lambda_k n^2 m^2}{s_k^3 b_k^3}$.

Summing up (36) from $k = 1$ to $K - 1$, we get

$$
0 \leq \mathbb{E}\left[ f(x^1) - f^* - \sum_{k=1}^{K-1} \frac{\eta_k}{2}\|\nabla f(x^k)\|^2 + \gamma_1(1-\lambda_1)^2\|v^0 - \nabla f(x^0)\|^2 \right.
$$

$$
+ \left(\frac{2\gamma_1 L^2}{s_1 b_1} + \alpha_1\left(1 - \frac{s_1 b_1}{nm}\right)\left(1 + \frac{2nm}{s_1 b_1}\right)L^2\right)\|x^1 - x^0\|^2
$$

$$
\left. + \left(\frac{2\gamma_1 \lambda_1^2}{s_1 b_1} + \alpha_1\left(1 - \frac{s_1 b_1}{nm}\right)\left(1 + \frac{s_1 b_1}{2nm}\right)\right)\frac{1}{nm} \sum_{i,j=1,1}^{n,m} \|\nabla f_{i,j}(x^0) - y_{i,j}^0\|^2 \right]. \quad (38)
$$

For $k = 0$, we directly uses (29), i.e.,

$$
\mathbb{E}[f(x^1) - f^*] \leq \mathbb{E}\left[ f(x^0) - f^* - \frac{\eta_0}{2}\|\nabla f(x^0)\|^2 - \left(\frac{1}{2\eta_0} - \frac{L}{2}\right)\|x^1 - x^0\|^2 + \frac{\eta_0}{2}\|v^0 - \nabla f(x^0)\|^2 \right].
$$
$$(39)$$

Now, we combine (38) and (39) to get

$$
\mathbb{E}\left[ \sum_{k=0}^{K-1} \frac{\eta_k}{2}\|\nabla f(x^k)\|^2 \right]
$$

$$
\leq \mathbb{E}\Big[f(x^0) - f^* + \big(\gamma_1(1-\lambda_1)^2 + \frac{\eta_0}{2}\big)\|v^0 - \nabla f(x^0)\|^2
$$

$$
+ \Big(\frac{2\gamma_1\lambda_1^2}{s_1 b_1} + \alpha_1\big(1 - \frac{s_1 b_1}{nm}\big)\big(1 + \frac{s_1 b_1}{2nm}\big)\Big)\frac{1}{nm}\sum_{i,j=1,1}^{n,m}\|\nabla f_{i,j}(x^0) - y_{i,j}^0\|^2\Big] \qquad (40)
$$

$$
\leq \mathbb{E}\Big[f(x^0) - f^* + \frac{\eta_0(1-\lambda_1(1-\lambda_1))}{2\lambda_1}\|v^0 - \nabla f(x^0)\|^2
$$

$$
+ \frac{2nm\lambda_1\eta_0}{s_1^2 b_1^2}\frac{1}{nm}\sum_{i,j=1,1}^{n,m}\|\nabla f_{i,j}(x^0) - y_{i,j}^0\|^2\Big] \qquad (41)
$$

$$
\leq f(x^0) - f^* + \frac{\eta_0}{2\lambda_1}\frac{nm - s_0 b_0}{(nm-1)s_0 b_0}\frac{1}{nm}\sum_{i,j=1,1}^{n,m}\|\nabla f_{i,j}(x^0)\|^2
$$

$$
+ \frac{2nm\lambda_1\eta_0}{s_1^2 b_1^2}\frac{nm - s_0 b_0}{nm}\frac{1}{nm}\sum_{i,j=1,1}^{n,m}\|\nabla f_{i,j}(x^0)\|^2 \qquad (42)
$$

$$
= f(x^0) - f^* + \frac{(nm - s_0 b_0)\eta_0\theta_0}{2nm s_0 b_0}G_0', \qquad (43)
$$

where (40) follows from the definition of $\eta_0$ in (37), (41) uses $\gamma_1 = \frac{\eta_0}{2\lambda_1}$ and $\alpha_1 = \frac{2nm\lambda_1\eta_0}{s_1^2 b_1^2}$, (42) uses $\lambda_0 = 1$, and (43) uses the definitions $\theta_0 := \frac{nm}{(nm-1)\lambda_1} + \frac{4nm\lambda_1 s_0 b_0}{s_1^2 b_1^2}$ and $G_0' := \frac{1}{nm}\sum_{i,j=1,1}^{n,m}\|\nabla f_{i,j}(x^0)\|^2$.

By randomly choosing $\widehat{x}^K$ from $\{x^k\}_{k=0}^{K-1}$ with probability $\eta_k / \sum_{t=0}^{K-1}\eta_t$ for $x^k$, (43) turns to

$$
\mathbb{E}[\|\nabla f(\widehat{x}^K)\|^2] \leq \frac{2(f(x^0) - f^*)}{\sum_{k=0}^{K-1}\eta_k} + \frac{(nm - s_0 b_0)\eta_0\theta_0 G_0'}{nm s_0 b_0 \sum_{k=0}^{K-1}\eta_k} \qquad (44)
$$

$\square$

## C.2 PROOFS OF COROLLARIES 3 AND 4

Now, we prove the detailed convergence results in Corollaries 3–4 with specific parameter settings.

***Proof of Corollary 3.*** First we know that (44) with $s_0 = n$ and $b_0 = m$ turns to

$$
\mathbb{E}[\|\nabla f(\widehat{x}^K)\|^2] \leq \frac{2(f(x^0) - f^*)}{\sum_{k=0}^{K-1}\eta_k}. \qquad (45)
$$

Then if we set $\lambda_k = \frac{s_k b_k}{2nm}$, $s_k \equiv \sqrt{n}$, and $b_k \equiv \sqrt{m}$ for any $k \geq 1$, then we know that $W_k := \frac{2}{\lambda_k s_k b_k} + \frac{8\lambda_k n^2 m^2}{b_k^3 s_k^3} \equiv 8$ and thus the stepsize $\eta_k \leq \frac{1}{L\left(1+\sqrt{W_{k+1}}\right)} \equiv \frac{1}{(1+\sqrt{8})L}$ for any $k \geq 0$. By plugging $\eta_k \leq \frac{1}{(1+\sqrt{8})L}$ into (45), we have

$$
\mathbb{E}[\|\nabla f(\widehat{x}^K)\|^2] \leq \frac{2(1+\sqrt{8})L(f(x^0) - f^*)}{K} = \epsilon^2,
$$

where the last equality holds by letting the number of iterations $K = \frac{2(1+\sqrt{8})L(f(x^0)-f^*)}{\epsilon^2}$. Thus the number of stochastic gradient computations for each client is

$$
\#\text{grad} = \sum_{k=0}^{K-1} b_k = m + \frac{(K-1)\sqrt{m}}{\sqrt{n}} \leq n + \sqrt{\frac{m}{n}}\frac{2(1+\sqrt{8})L(f(x^0) - f^*)}{\epsilon^2}.
$$

$\square$

***Proof of Corollary 4.*** First we recall (44) here:

$$
\mathbb{E}[\|\nabla f(\widehat{x}^K)\|^2] \leq \frac{2(f(x^0) - f^*)}{\sum_{k=0}^{K-1}\eta_k} + \frac{(nm - s_0 b_0)\eta_0\theta_0 G_0'}{nm s_0 b_0 \sum_{k=0}^{K-1}\eta_k}. \qquad (46)
$$

In this corollary, we do not compute any full gradients even for the initial point. We set the client sample size $s_k \equiv \sqrt{n}$ and minibatch size $b_k \equiv \sqrt{m}$ for any $k \geq 0$. So we need consider the second term of (46) since $s_0 b_0 = \sqrt{nm}$ is not equal to $nm$. Similar to Corollary 3, if we set $\lambda_k = \frac{s_k b_k}{2nm}$ for any $k \geq 1$, then we know that $W_k := \frac{2}{\lambda_k s_k b_k} + \frac{8\lambda_k n^2 m^2}{b_k^3 s_k^3} \equiv 8$ and thus the stepsize $\eta_k \leq \frac{1}{L\left(1+\sqrt{W_{k+1}}\right)} \equiv \frac{1}{(1+\sqrt{8})L}$ for any $k \geq 0$. Now, we can change (46) to

$$
\begin{aligned}
\mathbb{E}[\|\nabla f(\widehat{x}^K)\|^2] &\leq \frac{2(1+\sqrt{8})L(f(x^0)-f^*)}{K} + \frac{(nm-s_0 b_0)\theta_0 G_0'}{nms_0 b_0 K} \\
&\leq \frac{2(1+\sqrt{8})L(f(x^0)-f^*) + 4G_0'}{K} \\
&= \epsilon^2,
\end{aligned}
\tag{47}
$$

where (47) holds by plugging the initial values of the parameters into the last term, and the last equality holds by letting the number of iterations $K = \frac{2(1+\sqrt{8})L(f(x^0)-f^*)+4G_0'}{\epsilon^2}$. Thus the number of stochastic gradient computations for each client is

$$
\#\text{grad} = \sum_{k=0}^{K-1} b_k = \frac{K\sqrt{m}}{\sqrt{n}} = \sqrt{\frac{m}{n}} \frac{2(1+\sqrt{8})L(f(x^0)-f^*) + 4G_0'}{\epsilon^2} = O\left(\sqrt{\frac{m}{n}} \frac{L\Delta_0 + G_0'}{\epsilon^2}\right).
$$

Note that $\Delta_0 := f(x^0) - f^*$ where $f^* := \min_x f(x)$. $\qquad\square$

## D   FURTHER IMPROVEMENT FOR CONVERGENCE RESULTS

Note that all parameter settings, i.e., $\{\eta_k\}$, $\{b_k\}$ and $\{\lambda_k\}$ in ZeroSARAH for Corollaries 1–2, only require the values of $L$ and $n$, and $\{\eta_k\}$, $\{s_k\}$, $\{b_k\}$, $\{\lambda_k\}$ in D-ZeroSARAH for Corollaries 3–4 only require the values of $L$, $n$ and $m$, both are the same as all previous algorithms. If one further allows other values, e.g., $\epsilon$, $G_0$ or $\widehat{\Delta}_0$, for setting the initial $b_0$, then the gradient complexity can be further improved. See Appendices D.1 and D.2 for better results of ZeroSARAH and D-ZeroSARAH, respectively.

### D.1   BETTER RESULT FOR ZeroSARAH

**Corollary 5** *Suppose that Assumption 1 holds. Choose stepsize $\eta_k \leq \frac{1}{(1+\sqrt{8})L}$ for any $k \geq 0$, minibatch size $b_k \equiv \sqrt{n}$ and parameter $\lambda_k = \frac{b_k}{2n}$ for any $k \geq 1$. Moreover, let $b_0 = \min\left\{\sqrt{\frac{nG_0}{\epsilon^2}}, n\right\}$ and $\lambda_0 = 1$. Then ZeroSARAH (Algorithm 2) can find an $\epsilon$-approximate solution for problem (1) such that*

$$
\mathbb{E}[\|\nabla f(\widehat{x}^K)\|^2] \leq \epsilon^2
$$

*and the number of stochastic gradient computations can be bounded by*

$$
\#\text{grad} = O\left(\sqrt{n}\left(\frac{L\Delta_0}{\epsilon^2} + \min\left\{\sqrt{\frac{G_0}{\epsilon^2}}, \sqrt{n}\right\}\right)\right).
$$

*Similarly, $G_0$ can be bounded by $G_0 \leq 2L\widehat{\Delta}_0$ via Assumption 1. Let $b_0 = \min\left\{\sqrt{\frac{nL\widehat{\Delta}_0}{\epsilon^2}}, n\right\}$, then we also have*

$$
\#\text{grad} = O\left(\sqrt{n}\left(\frac{L\Delta_0}{\epsilon^2} + \min\left\{\sqrt{\frac{L\widehat{\Delta}_0}{\epsilon^2}}, \sqrt{n}\right\}\right)\right).
$$

**Remark:** The result of Corollary 5 for ZeroSARAH is the best one compared with Corollaries 1–2. In particular, it recovers Corollary 1 when $b_0 = n$. In the case $b_0 < n$ (never computes any full gradients even for the initial point), then $\#\text{grad} = O\left(\sqrt{n}\left(\frac{L\Delta_0}{\epsilon^2} + \sqrt{\frac{G_0}{\epsilon^2}}\right)\right)$ which is better than the result $O\left(\frac{\sqrt{n}(L\Delta_0+G_0)}{\epsilon^2}\right)$ in Corollary 2. Similar to the Remark after Corollary 2, if we consider $L$,

$\Delta_0$, $G_0$ or $\widehat{\Delta}_0$ as constant values then the stochastic gradient complexity in Corollary 5 is $\#\text{grad} = O(\frac{\sqrt{n}}{\epsilon^2})$, i.e., full gradient computations do not appear in ZeroSARAH and the term '$n$' also does not appear in its convergence result. If we further assume that loss functions $f_i$'s are non-negative, i.e., $\forall x, f_i(x) \geq 0$ (usually the case in practice), we can simply bound $\widehat{\Delta}_0 := f(x^0) - \widehat{f}^* \leq f(x^0)$ and then $b_0$ can be set as $\min\left\{\sqrt{\frac{nLf(x^0)}{\epsilon^2}}, n\right\}$ for Corollary 5.

***Proof of Corollary 5.*** First we recall (25) here:

$$\mathbb{E}[\|\nabla f(\widehat{x}^K)\|^2] \leq \frac{2(f(x^0) - f^*)}{\sum_{k=0}^{K-1} \eta_k} + \frac{(n - b_0)(4\gamma_0 + 2\alpha_0 b_0)}{nb_0 \sum_{k=0}^{K-1} \eta_k} \frac{1}{n}\sum_{j=1}^{n} \|\nabla f_j(x^0)\|^2. \quad (48)$$

Note that here we also need consider the second term of (48) since $b_0$ may be less than $n$. Similar to Corollary 2, if we set $\lambda_k = \frac{b_k}{2n}$ and $b_k \equiv \sqrt{n}$ for any $k \geq 1$, then we know that $M_k := \frac{2}{\lambda_k b_k} + \frac{8\lambda_k n^2}{b_k^3} \equiv 8$ and thus the stepsize $\eta_k \leq \frac{1}{L(1+\sqrt{M_{k+1}})} \equiv \frac{1}{(1+\sqrt{8})L}$ for any $k \geq 0$. For the second term, we recall that $\gamma_0 \geq \frac{\eta_0}{2\lambda_1} = \frac{\sqrt{n}}{(1+\sqrt{8})L}$ and $\alpha_0 \geq \frac{2n\lambda_1\eta_0}{b_1^2} = \frac{1}{(1+\sqrt{8})L\sqrt{n}}$. It is easy to see that $\gamma_0 \geq \alpha_0 b_0$ since $b_0 \leq n$. Now, we can change (48) to

$$\mathbb{E}[\|\nabla f(\widehat{x}^K)\|^2] \leq \frac{2(1+\sqrt{8})L(f(x^0) - f^*)}{K} + \frac{6(n - b_0)}{\sqrt{n}b_0 K} \frac{1}{n}\sum_{j=1}^{n} \|\nabla f_j(x^0)\|^2$$

$$= \frac{2(1+\sqrt{8})L(f(x^0) - f^*)}{K} + \frac{6(n - b_0)G_0}{\sqrt{n}b_0 K} \quad (49)$$

$$= \epsilon^2,$$

where (49) is due to the definition $G_0 := \frac{1}{n}\sum_{i=1}^{n} \|\nabla f_i(x^0)\|^2$, and the last equality holds by letting the number of iterations $K = \frac{2(1+\sqrt{8})L(f(x^0)-f^*)}{\epsilon^2} + \frac{6(n-b_0)G_0}{\sqrt{n}b_0\epsilon^2}$. Thus the number of stochastic gradient computations is

$$\#\text{grad} = \sum_{k=0}^{K-1} b_k = b_0 + \sum_{k=1}^{K-1} b_k$$

$$= b_0 + (K-1)\sqrt{n} \leq b_0 + \frac{2(1+\sqrt{8})\sqrt{n}L(f(x^0) - f^*)}{\epsilon^2} + \frac{6(n - b_0)G_0}{b_0\epsilon^2}.$$

By choosing $b_0 = \min\{\sqrt{\frac{nG_0}{\epsilon^2}}, n\}$, we have

$$\#\text{grad} \leq \sqrt{n}\left(\frac{2(1+\sqrt{8})L(f(x^0) - f^*)}{\epsilon^2} + \min\left\{7\sqrt{\frac{G_0}{\epsilon^2}}, \sqrt{n}\right\}\right)$$

$$= O\left(\sqrt{n}\left(\frac{L\Delta_0}{\epsilon^2} + \min\left\{\sqrt{\frac{G_0}{\epsilon^2}}, \sqrt{n}\right\}\right)\right).$$

Similarly, $G_0$ can be bounded by $G_0 \leq 2L(f(x^0) - \widehat{f}^*)$ via Assumption 1 and let $b_0 = \min\{\sqrt{\frac{nL\widehat{\Delta}_0}{\epsilon^2}}, n\}$, then we have

$$\#\text{grad} = O\left(\sqrt{n}\left(\frac{L\Delta_0}{\epsilon^2} + \min\left\{\sqrt{\frac{L\widehat{\Delta}_0}{\epsilon^2}}, \sqrt{n}\right\}\right)\right).$$

Note that $\Delta_0 := f(x^0) - f^*$, where $f^* := \min_x f(x)$, and $\widehat{\Delta}_0 := f(x^0) - \widehat{f}^*$, where $\widehat{f}^* := \frac{1}{n}\sum_{i=1}^{n} \min_x f_i(x)$. $\square$

## D.2 BETTER RESULT FOR D-ZeroSARAH

**Corollary 6** *Suppose that Assumption 2 holds. Choose stepsize $\eta_k \leq \frac{1}{(1+\sqrt{8})L}$ for any $k \geq 0$, clients subset size $s_k \equiv \sqrt{n}$, minibatch size $b_k \equiv \sqrt{m}$ and parameter $\lambda_k = \frac{s_k b_k}{2nm}$ for any $k \geq 1$.*

*Moreover, let $s_0 = \min\left\{\sqrt{\frac{nG_0'}{m\epsilon^2}}, n\right\}$ and $b_0 = m$ (or $b_0 = \min\left\{\sqrt{\frac{mG_0'}{n\epsilon^2}}, m\right\}$ and $s_0 = n$), and $\lambda_0 = 1$. Then* D-ZeroSARAH *(Algorithm 3) can find an $\epsilon$-approximate solution for distributed problem* (2) *such that*

$$\mathbb{E}[\|\nabla f(\widehat{x}^K)\|^2] \le \epsilon^2$$

*and the number of stochastic gradient computations for each client can be bounded by*

$$\#\text{grad} = O\left(\sqrt{\frac{m}{n}}\left(\frac{L\Delta_0}{\epsilon^2} + \min\left\{\sqrt{\frac{G_0'}{\epsilon^2}}, \sqrt{nm}\right\}\right)\right).$$

***Proof of Corollary 6.*** First we recall (44) here:

$$\mathbb{E}[\|\nabla f(\widehat{x}^K)\|^2] \le \frac{2(f(x^0) - f^*)}{\sum_{k=0}^{K-1} \eta_k} + \frac{(nm - s_0b_0)\eta_0\theta_0 G_0'}{nms_0b_0 \sum_{k=0}^{K-1} \eta_k}. \tag{50}$$

Similar to Corollary 4, here we also need consider the second term of (50) since $s_0b_0$ may be less than $nm$. Similarly, if we set $\lambda_k = \frac{s_kb_k}{2nm}$, $s_k \equiv \sqrt{n}$, and $b_k \equiv \sqrt{n}$ for any $k \ge 1$, then we know that $W_k := \frac{2}{\lambda_k s_k b_k} + \frac{8\lambda_k n^2 m^2}{b_k^3 s_k^3} \equiv 8$ and thus the stepsize $\eta_k \le \frac{1}{L\left(1+\sqrt{W_{k+1}}\right)} \equiv \frac{1}{(1+\sqrt{8})L}$ for any $k \ge 0$. Then (50) changes to

$$\begin{aligned}
\mathbb{E}[\|\nabla f(\widehat{x}^K)\|^2] &\le \frac{2(1+\sqrt{8})L(f(x^0) - f^*)}{K} + \frac{(nm - s_0b_0)\theta_0 G_0'}{nms_0b_0 K} \\
&= \frac{2(1+\sqrt{8})L(f(x^0) - f^*)}{K} + \frac{6(nm - s_0b_0)G_0'}{\sqrt{nm}s_0b_0 K} \\
&= \epsilon^2,
\end{aligned} \tag{51}$$

where (51) by figuring out $\theta_0$ with the initial values of the parameters, and the last equality holds by letting the number of iterations $K = \frac{2(1+\sqrt{8})L(f(x^0)-f^*)}{\epsilon^2} + \frac{6(nm-s_0b_0)G_0'}{\sqrt{nm}s_0b_0\epsilon^2}$. Thus the number of stochastic gradient computations for each client is

$$\begin{aligned}
\#\text{grad} = \sum_{k=0}^{K-1} b_k &= \frac{s_0}{n}b_0 + \frac{(K-1)\sqrt{m}}{\sqrt{n}} \\
&\le \sqrt{\frac{m}{n}}\frac{2(1+\sqrt{8})L(f(x^0)-f^*)}{\epsilon^2} + \frac{s_0b_0}{n} + \frac{6(nm-s_0b_0)G_0'}{ns_0b_0\epsilon^2} \\
&\le \sqrt{\frac{m}{n}}\left(\frac{2(1+\sqrt{8})L(f(x^0)-f^*)}{\epsilon^2} + \min\left\{7\sqrt{\frac{G_0'}{\epsilon^2}}, \sqrt{nm}\right\}\right) \\
&= O\left(\sqrt{\frac{m}{n}}\left(\frac{L\Delta_0}{\epsilon^2} + \min\left\{\sqrt{\frac{G_0'}{\epsilon^2}}, \sqrt{nm}\right\}\right)\right),
\end{aligned} \tag{52}$$

where (52) holds by choosing $s_0b_0 = \min\left\{\sqrt{\frac{nmG_0'}{\epsilon^2}}, nm\right\}$. It can be satisfied by letting $s_0 = \min\left\{\sqrt{\frac{nG_0'}{m\epsilon^2}}, n\right\}$ and $b_0 = m$ (or $s_0 = n$ and $b_0 = \min\left\{\sqrt{\frac{mG_0'}{n\epsilon^2}}, m\right\}$). The last equation uses the definition $\Delta_0 := f(x^0) - f^*$ where $f^* := \min_x f(x)$. $\square$

