# OpenReview forum: "ZeroSARAH: Efficient Nonconvex Finite-Sum Optimization with Zero Full Gradient Computations"
_ICLR.cc/2022/Conference — ICLR 2022 Submitted_

### Official Review · Reviewer_YmLP · 2021-10-30

**Correctness:** 4
**Technical Novelty And Significance:** 3
**Empirical Novelty And Significance:** 3
**Recommendation:** 8
**Confidence:** 2

**Main Review:**

Overall it's a nice method and I don't see anything wrong with the proofs (though I really only skimmed). My main counterthought is that the paper didn't really address SAG and SAGA, which are other mainstream VR methods that also do not require full gradient calls, do not even require full minibatch computations, and also have competitive convergence rates. The downside of SAG and SAGA seem to be the same as this method, which is the high memory cost; the need to store the yi's seems to be similar to the issues of SAG and SAGA. As it stands, having such a high memory cost in general makes these methods prohibitive to most deep learning applications, and is on the same order of "issues" as SVRG.

The other (maybe not that minor) thing is that, the authors say that although minibatch versions of SVRG and such methods exist, their convergence depends on bounding the variance of the stochastic gradients, which it is true is hard in general. However, this is necessary for the proofs of ALL VR methods, not just the minibatched versions. It seems like the authors bypassed this issue via telescoping, so perhaps it just isn't an issue with SARAH. But I don't believe this is somehow a new issue with minibatching to estimate the stale full gradient, is my point.

**Summary Of The Paper:**

This paper introduces a form of SARAH (the variance reduction method)  that avoids taking full gradients, which is called ZeroSARAH, and also a distributed version (D-ZeroSARAH). They offer convergence results and shows that it is on the same order of gradient calls to $$\epsilon$$-gradient norm as other state-of-the-art variance reduction methods, but claim it is without ever requiring full gradients.



**Summary Of The Review:**

See above

---

> ### Author Response · Authors · 2021-11-23
> **Thanks for your positive evaluation**
>
> Thank you for the insightful comments and positive evaluation.
>
> - Regarding the memory cost: **Yes, it is a trade-off (memory cost vs. no full gradient) for the proposed ZeroSARAH method**. This is also the motivation of the second part (distributed/federated setting) of our paper, the drawback of memory cost is alleviated/avoided in the distributed setting, i.e., the memory cost is distributed among all the clients.
>
> - However, **full gradient computation is the main issue in distributed/federated learning since full participation with all clients/devices usually is impractical.** Note that the key challenge we focus on in our paper is to resolve the issue of full gradient computations. We do not expect that we can solve that issue for free with no cost. Every algorithm has its own applicable situations and trade-offs.
>
> Thanks again!

---

> > ### Comment · Reviewer_YmLP · 2021-11-28
> > **thanks for response, but other reviewers' concerns are worth thinking about**
> >
> > I do think that in the case of computation / memory tradeoff, proper reference and comparison with SAG and SAGA is important, because this idea is not as novel as the paper currently presents. It is possible that the convergence rates for ZeroSARAH is more powerful than SAG/SAGA, but that point doesn't seem as highlighted in the main paper,  which does not mention SAG and SAGA. I think this kind of discussion, and in fact a discussion of which regimes ZeroSARAH is competitive over SAG and SAGA in practical experiments, would significantly strengthen the paper.

---

> > > ### Author Response · Authors · 2021-11-28
> > > **To Reviewer YmLP**
> > >
> > > Thanks for your helpful feedback.
> > > **Yes. Our ZeroSARAH achieves much better convergence results than SAGA.**  The result of nonconvex SAGA can be found in e.g. [1].
> > >
> > > As we pointed out in the Abstract and Introduction:
> > > > Note that SVRG, SAGA and their variants typically achieve weaker convergence results than variants of SARAH: $\frac{n^{2/3}}{\epsilon^2}$ vs. $\frac{\sqrt{n}}{\epsilon^2}$. Thus we focus on the variant of SARAH.
> > >
> > > **The main contribution of our work is to remove the full gradient computations and achieve the state-of-the-art results (i.e., $\frac{\sqrt{n}}{\epsilon^2}$) *simultaneously*, unlike the weaker result $\frac{n^{2/3}}{\epsilon^2}$ in SAGA.**
> > >
> > > Thanks for your constructive comments. We will add more discussions/comparisons between SAGA and our ZeroSARAH in the next version.
> > >
> > >
> > >
> > > [1] Zhize Li and Peter Richtarik. A unified analysis of stochastic gradient methods for nonconvex federated optimization. *arXiv preprint arXiv:2006.07013*, 2020.

---

### Official Review · Reviewer_xTXr · 2021-11-02

**Correctness:** 3
**Technical Novelty And Significance:** 3
**Empirical Novelty And Significance:** 2
**Recommendation:** 5
**Confidence:** 3

**Main Review:**

The paper first presents ZeroSARAH, which slightly modifies SARAH in the sense that ZeroSARAH replaces the past gradient estimator $v^{k-1}$ of SARAH with the weighted sum of $v^{k-1}$ and another estimator to avoid the periodic computation of the full gradient. The technique is definitely new, but still a bit incremental.


In the centralized setting (ZeroSARAH), the authors establish the ZeroSARAH's complexity of $O(n+\frac{\sqrt{n} L \Delta_0}{\epsilon^2})$ or $O(\frac{\sqrt{n} (L \Delta_0+G_0)}{\epsilon^2})$ versus $O(n+\frac{\sqrt{n} L \Delta_0}{\epsilon^2})$ of SARAH. The term $n$ in SARAH just means that you must run at least one iteration of the algorithm [1], which takes the full gradient snapshot, and is usually dominated by the other term due to the large condition numbers in practice. Therefore, the complexity achieved by ZeroSARAH is essentially comparable to SARAH's, where $O(\frac{\sqrt{n} (L \Delta_0+G_0)}{\epsilon^2})$ can be seen as the interpretation of the fact that no full gradient computation is needed. However, in this centralized setting, avoiding full gradient computation does not provide any real benefit. ZeroSARAH still serves as the basis for the development of D-ZeroSARAH for distributed setting where avoidance of full gradient computation can be beneficial.


In the distributed setting (D-ZeroSARAH), the complexities established for D-ZeroSARAH are comparable to distributed Sarah, which follows the same reasons as above. The main benefit of D-ZeroSARAH is the fact that it does not require full gradient computation, which can alleviate the burden of synchronizing all clients. This is an interesting perspective that has not been widely considered yet. Nevertheless, since both D-ZeroSARAH and distributed Sarah have the same gradient complexity $O(\sqrt{\frac{m}{n}} \frac{1}{\epsilon^2})$ and D-ZeroSARAH requires less gradient evaluations per iterations, the number of iterations, i.e. communication rounds, required by D-ZeroSARAH is more than by SARAH. The communication complexity (i.e. communication rounds) is one of the most common metrics considered by the distributed optimization literature, which also corresponds to the fact that setting up a synchronization round is expensive in practice. All in all, D-ZeroSARAH can be viewed as a variant of distributed SARAH with reasonable tradeoffs in number of gradient computations per iteration and communication complexity. Perhaps the authors can add some discussion on communication complexity.

Extensive experiments for both of ZeroSARAH and D-ZeroSARAH were conducted to verify the theories and robustness of the proposed methods.


Typos/Mismatch: In Corollary 3, the parameter is set such that $b_k= \sqrt{m}$, but in its proof it seems that $b_k=\sqrt{m/n}$.





[1] Pham, Nhan H., Lam M. Nguyen, D. Phan and Quoc Tran-Dinh. “ProxSARAH: An Efficient Algorithmic Framework for Stochastic Composite Nonconvex Optimization.” ArXiv abs/1902.05679 (2020).

**Summary Of The Paper:**

The paper proposes ZeroSARAH (and its distributed version D-ZeroSARAH), a variant of the well-known variance-reduced method SARAH, for non convex finite-sum optimization. The main benefit of ZeroSARAH and D-ZeroSARAH is that they do not require any full gradient computations, in contrast to other known variance reduction algorithms. In distributed setting, this can avoid the burden of synchronizing all clients.

**Summary Of The Review:**

Overall, the paper proposes ZeroSARAH and its distributed version D-ZeroSARAH, as variants of SARAH that avoid full gradient computation. Under the distributed setting (where such feature can be beneficial), requiring less number of gradient computations per iteration means that D-ZeroSARAH requires more iterations, i.e. communication rounds, than distributed SARAH to converge, since the  gradient complexity of these 2 algorithms are asymptotically the same. This provides some interesting tradeoffs, but in general, communication complexity is usually more prioritized in practice. D-ZeroSARAH can address certain situations where synchronizing all clients is impractical. This is a new perspective for this class of problems and can facilitate future work in this direction. The proposed methods are empirically validated.

---

> ### Author Response · Authors · 2021-11-23
> **Clarifications of misunderstandings**
>
> Thanks for the positive evaluation of our work, we appreciate it.
>
> We noticed a  misunderstanding wrt the communication round/complexity of D-ZeroSARAH vs. that of Distributed SARAH. Let us elaborate:
>
>
> - **Our D-ZeroSARAH method uses the same umber of communication rounds as Distributed SARAH**, which is similar to the standard setting of ZeroSARAH vs. SARAH where both use the same number of iterations.
>
> - **There is no trade-off between the number of communication rounds and full gradient computations here.** We would like to point out that (distributed) SARAH does not always compute the full gradient for each iteration (do full participation for each communication round), they do that periodically.
>
>
> We hope our response is helpful. If clarifying the above point was important to you, please do consider raising your score. Thank you!

---

> > ### Comment · Reviewer_xTXr · 2021-11-28
> > **Discussion**
> >
> > From my previous review, I did not think that ZeroSARAH's complexity is better than, and at most only comparable to the state-of-the-art complexity. However, since you strongly assert and defend your point that your algorithm improves the complexity in certain regimes throughout not only the paper but also the rebuttal, firm justification of such regimes must be made. Otherwise, the statement would be misleading. The characterization of your regimes that $G_0$ or $\Delta_0$ can be small in response to other reviewers is not solid enough. Further meaningful insights of your regimes' practicality, or at least some empirical verification of $G_0$ or $\Delta_0$ are necessary. I would only suggest weak acceptance if major revisions on the result interpretation are made.

---

> > > ### Author Response · Authors · 2021-11-28
> > > **To Reviewer xTXr**
> > >
> > > **We always emphasize that our main contributions focus on removing periodical full gradient computations.** Note that in distributed/federated learning, periodic computation of full gradient needs to periodically synchronize all clients/devices, which usually is impractical (e.g., partial participation is a key feature/requirement for federated learning).  Moreover, we have already successfully applied our ZeroSARAH (due to its feature of avoiding full participation) into our following federated learning projects and achieved the desired results. **We do not focus on improving previous convergence results (so we do not care how small the improvements are), but we still want to emphasize that *our results can at least match previous best results* (see our Corollaries 1 and 3 in Tables 1 and 2).**
> > >
> > > According to our last response to Reviewer qPn9, we also have already emphasized this point.
> > > > We emphasize that our **Corollary 1 and Corollary 3** (both do not involve the  $G_0$ or $\hat{\Delta}_0$) only compute the full gradient **once** at the initial point and **obtain the best-known results**, while **previous algorithms need to compute full gradients periodically** (please see our Table 1 and 2).
> > > For our Corollaries 2 and 4, they only improve our Corollaries 1 and 3 by further not computing the full gradient at the initial point (thus these two results involve the parameter $G_0$ or $\hat{\Delta}_0$). However, from the result view, we already said that the improvement regime of Corollaries 2 and 4 vs. Corollaries 1 and 3 can be very small. From the algorithm view, Corollaries 2 and 4 only improve *one* full gradient computation vs. Corollaries 1 and 3, while our Corollaries 1 and 3 significantly improve previous works by removing periodical full gradient computations. **Thus we would conclude that our Corollaries 1 and 3 make the main contributions in our current paper.**
> > > **That means even if we only present these two corollaries in our work is enough for showing the benefit of removing periodical full gradient computations.**
> > >
> > >
> > > Your comments seem to ignore our main contributions and focus on minor points. Please judge our paper according to the key contributions.
> > >
> > > Authors

---

### Official Review · Reviewer_qPn9 · 2021-11-02

**Correctness:** 3
**Technical Novelty And Significance:** 3
**Empirical Novelty And Significance:** 2
**Recommendation:** 5
**Confidence:** 4

**Main Review:**

1. The authors say in the abstract, intro and conclusion that "their complexity improves existing ones in current regimes", but I cannot see this written down explicitly. The closest I can find is the Remark after Corol. 2 that states "if G_0, \hat \Delta_0" are constants, which is obviously a too much of a simplification and also not true in most cases. For example, why is $\hat \Delta_0$ is even finite? Assuming the objective is lower bounded is a common assumption, but of course each component can be without a lower bound (take linear functions. Their sum can lower bounded whereas they individually are not). Then what will happen to $\hat \Delta_0$?

2. Due to my previous point, it is not clear if it is possible to give a fair comparison between the current complexity and previous best complexity. The new complexities depend on $G_0, \hat \Delta_0$ and their relation to the other terms is quite unclear.

3. In the experiments, only comparison is with SARAH which is insufficient. Since one of the main motivations of the paper is distributed optimization, I think the authors need to compare with the algorithms from Table 2, for example SCAFFOLD, or other SOTA algorithms in this setting to see if the new method is really useful in practice.

4. Please define $\hat x_K$ either in the algorithm or the theorem/corollaries. I had to check the proof to see how it is defined.

I am of course open to increase my score if my concerns in 1,2,3 are addressed.

**Summary Of The Paper:**

This paper introduce ZeroSARAH which is a variant of variance reduced SARAH algorithm, for solving $\min_x 1/n \sum_{i=1}^n f_i(x)$ with nonconvex and smooth $f_i$. ZeroSARAH and its distributed version are the first algorithms that do not require any full gradient computations. Most algorithms for this setting either required computing the full gradient once in the beginning or periodically. The paper also argues that in distributed optimization, computation of full gradients can be a bottleneck and therefore avoiding it is important.


**Summary Of The Review:**

Overall, I agree that an algorithm with zero full gradient computation is definitely interesting, it is not clear if this brings any advantage in theory (due to different parameters in the bounds) and also in practice (experiments are only compared to SARAH and not with the SOTA solvers in distributed settings.)

---

> ### Author Response · Authors · 2021-11-23
> **Response to all concerns raised by Reviewer qPn9**
>
> **Q1.** Regarding our results vs. previous results.
>
> **A1.** First, we point out that the original sentence in our paper was **”our results can improve existing ones in *certain* regimes“** not **”$\ldots$ improve existing ones in *current* regimes“**. Note that our results do improve previous best results for certain regimes when $G_0$ is small, e.g., the gradients at initial point $x^0$ are very close (Please see our Table 1 and 2). We will add a clarification of this to other parts of the paper as well so that this point can't be missed. However, please note that **there is nothing with wrong with our statement - it is factually correct. As such, we think it should not be used to decrease the score of our work.** Thanks!
>
> **Q2.** Regarding the $G_0$ or $\hat{\Delta}_0$.
>
> **A2.**
>
> - As we discussed above, $G_0$ can be small, e.g., when the gradients at initial point $x^0$ are very close. For $\hat{\Delta}_0$, we would like to mention that it is usually not very large. In particular, it is almost the same as $\Delta_0$ for overparameterized networks/problems. Also, the individual loss functions are usually lower bounded by 0 since most widely-used loss functions are indeed nonnegative. This is really not a problem.
>
> - Moreover, we would simply refer people to our Corollary 1 (which just computes the full gradient once at the initial point, while previous algorithms need to compute full gradients periodically) if you are very worried about $G_0$ or $\hat{\Delta}_0$ since in this case these terms do not appear in the final result!
>
>
> **Q3.** Regarding the experiments
>
> **A3.**
>
> - We would suggest the reviewer to look at Table 2 more carefully. First, **previous algorithms do not achieve SOTA results except for distributed SARAH-type methods**, and importantly, they focus on different problems and settings, e.g., decentralized/compressed/local updated settings (see the footnotes of Table 2). Thus, it is fine that we compare ZeroSARAH with SARAH in the experiments. This is the most relevant and meaningful comparison.
>
> - We will add a brief explanation of this in the paper. We believe this is a very minor point.
>
>
> ---
>
> We would very much appreciate if the reviewer could re-evaluate our work in the light of the above response. Thank you!

---

> > ### Comment · Reviewer_qPn9 · 2021-11-26
> > **Discussion**
> >
> > - The complexity comparisons between the new algorithm with 0 gradient computation vs existing results is not fair. The authors did not address this point in their rebuttal, but said "we did not say we improve the complexity, we only said we *can* improve the complexity in certain regimes", which is not satisfactory for me. The authors need to provide a clear comparison and make it clear when their method is better/worse than existing methods rather than such vague comments which don't say much. The authors say "there exist some regimes (no matter how small they are), our results can outperform previous results". However, I argue that when the **regime** that the authors mention is "too small", a paper does not necessarily warrant publication at a top venue.
> >
> > The authors claim the new constants in their bounds *can be* small in applications, but they are not small even for linear functions.
> >
> > - It seems the algorithm which computes 1 full gradient can match the best complexities (and compute full gradient once, compared to the existing ones that compute periodically), I thank the authors for the clarification. However based on the concerns on storage as raised by other reviewers, it is not clear to me if this is a sufficient contribution for publication, I think a more detailed comparison is needed.
> >
> > - The authors' response for the experiments is also not satisfactory, which is essentially saying that "our comparison is fine". I am not convinced on how the new method will compare with other methods in Table 2, which are for distributed setting and the response above  don't seem to be trying to clarify that. It is not an excuse that the previous methods are on slightly different settings, authors should have done the work to find a setting to compare with the other algorithms in Tab. 2.
> >
> > Therefore, my opinion remains negative for the paper at this point.

---

> > > ### Author Response · Authors · 2021-11-26
> > > **To Reviewer qPn9**
> > >
> > > Thanks for your feedback. **However, similar to our responses to Reviewer bVt3, we also do not agree with your opinions above.**
> > >
> > > We have already replied to you in our original response **"Moreover, we would simply refer people to our Corollary 1 (which just computes the full gradient once at the initial point, while previous algorithms need to compute full gradients periodically) if you are very worried about $G_0$ or $\hat{\Delta}_0$ since in this case these terms do not appear in the final result!"**.
> > > We emphasize that our **Corollary 1 and Corollary 3** only compute the full gradient **once** at the initial point and **obtain the best-known results**, while **previous algorithms need to compute full gradients periodically** (please see our Table 1 and 2).
> > > For our Corollaries 2 and 4, they only improve our Corollaries 1 and 3 by further not computing the full gradient at the initial point (thus these results involve the parameter $G_0$ or $\hat{\Delta}_0$). However, from the result view, we already said that the improvement regime of Corollaries 2 and 4 vs. Corollaries 1 and 3 can be very small. From the algorithm view, Corollaries 2 and 4 only improve *one* full gradient computation vs. Corollaries 1 and 3, while our Corollaries 1 and 3 significantly improve previous works by removing periodical full gradient computations. **Thus we would conclude that our Corollaries 1 and 3 make the main contributions in our current paper.**
> > > **That means even if we only present these two corollaries in our work is enough for showing the benefit of removing periodical full gradient computations.**
> > > *Note that in distributed/federated learning, periodic computation of full gradient needs to periodically synchronize all clients/devices, which usually is impractical (e.g., partial participation is a key feature/requirement for federated learning).*  Moreover, we have already successfully applied our ZeroSARAH (due to its feature of avoiding full participation) into our following federated learning projects and achieved the desired results.
> > >
> > > Regarding the experiments, we always emphasize that our main contributions focus on removing impractical full gradient computations. We do not think it is necessary for us to compare with **other algorithms given in other settings** e.g., decentralized/local updated/compressed settings in Table 2 (note that these algorithms also **do not obtain the state-of-the-art results** compared with distributed SARAH-type algorithm that we have already compared in the experiments). **Also, we do not think this minor point can affect the main contributions (avoiding full gradient computation/full participation in distributed/federated learning) of our work.**
> > >
> > >
> > > **In sum, we think your comments do not do a reasonable judgment for our contributions. We kindly request the reviewer could re-evaluate our work.**
> > >
> > > Authors

---

### Official Review · Reviewer_bVt3 · 2021-11-03

**Correctness:** 2
**Technical Novelty And Significance:** 3
**Empirical Novelty And Significance:** 2
**Recommendation:** 3
**Confidence:** 5

**Main Review:**

- The authors propose ZeroSARAH and its variant for distributed framework. They highlight that "Avoiding any full gradient computations is important especially in this distributed setting, periodic computation of full gradient across all n clients may be impossible or unaffordable." Furthermore, "the main advantage of (their) algorithms is that they do not need to compute any full gradients".

- However, ZeroSARAH needs a group of auxiliary variables $y_i^k$ for $i= 1,\dots, n$ and $k = 0,1,2,\dots$. I find it surprising when the authors did not discuss these variables in the description of Algorithm 2 (Section 4.1). The update in line 5 in fact requires us to compute the summation of $n$ such variables $\sum_{j=1}^n y_j^{k-1}$, which is relatively similar to computing the full gradient. Again, I am surprised that the authors did not discuss the storage cost needed for ZeroSARAH. From my understanding, this algorithm needs to keep a table of past gradients in the auxiliary variables $y_i^k$, and at the moment I can not find a solution to get rid of this storage cost. For that reason, it is reasonable that your algorithm does not need many full gradient computations while still achieves the same state-of-the-art rate. I recommend that the authors discuss your method with SAG/SAGA where these methods also need to store a table of past gradients.

- Similarly in the distributed setting, although we don't need a full gradient, we still need to compute a sum of $n$ variables (Line 11 of Algorithm 3). Hence this may slow down the training process no matter the fact that we don't collect every gradient.

- For these reasons above, I do not see that ZeroSARAH has any major advantage over existing works. Please note that not every VR method needs to compute full gradients. For example, the reference [Inexact SARAH Algorithm for Stochastic Optimization] (Nguyen et al, 2020) only compute a mini-batch gradient in the training process. This should be added and discussed in this paper.

- The authors might want to edit some statements that could be confusing/misleading. For example, the statement "all existing variance-reduced methods, including SARAH, SVRG, SAGA and their variants, need to compute the full gradient" while "SCSG (Lei et al., 2017), SVRG+ (Li & Li, 2018), PAGE (Li et al., 2021)) may avoid full gradient computations by using a large minibatch of stochastic gradients instead". Also the statement "(our methods) improve the previous best-known result" is also misleading.

- A minor point: the authors might want to specify the output $\hat{x}^K$ in the main paper.

**Summary Of The Paper:**

The authors propose a variance-reduced algorithm ZeroSARAH for solving the nonconvex finite-sum problems. They also propose a distributed variant D-ZeroSARAH for the corresponding distributed framework. They provide the state-of-the-art convergence results for these methods using standard assumptions. The authors claim that both methods does not require computing full gradients, which gives a major advantage when the number of samples $n$ is large. They further validate ZeroSARAH using nonconvex linear regression and binary classification for neural networks.

**Summary Of The Review:**

The authors present a new method named ZeroSARAH where the main advantage is that they do not need to compute any full gradients. However this algorithms may requires more storage/computational cost than the previous methods that use minibatch to update. Therefore I do not see any advantage and do not support publication at the moment when these issues are unresolved.

---

> ### Author Response · Authors · 2021-11-23
> **Response to all comments of Reviewer bVt3**
>
> Thank you for your time and effort. While your self-reported confidence is high, we wish to point out that your review and score is based on several key misunderstandings of our results. We would be most happy to clarify, and hope that these clarifications will allow you to re-evaluate your judgement. Thanks in advance!
>
> **Q1.** Regarding the auxiliary variables $\{y_i^k\}$.
>
> **A1.** It seems that you believe that our method needs to store all vectors $y_i^k$ for all nodes $i=1, 2, \ldots, n$ and  all iterates $k=0, 1, 2, \ldots$. **This is not the case.** The situation here is similar to the situation in classical SAG/SAGA. That is, we only need to remember one vector $y_i$ for each node $i=1, 2, \ldots, n$. That is, $y_i^k$ for different iteration $k$ reuses the same variable $y_i$. The superscript $k$ is just used for clear algorithmic presentation and to facilitate the proof. This is similar to the standard notation of the type $x^0, x^1, \ldots, x^k$ for the iterates, which does not mean that it is necessary to store all history points. Only the last point is stored. **So, in summary, our method requires only one vector to be stored per node, which is a very reasonable cost.**
>
> **Q2.** Regarding the summation of $n$ variables: $\sum_{j=1}^n y_j^k$.
>
> **A2.** Similarly to the SAG/SAGA situation, it is not efficient to compute this summation naively/explicitly. One only needs to use a variable to denote the result of this summation, e.g., let $P^k:= \sum_{j=1}^n y_j^k$, and it can be efficiently updated via $P^{K} = P^{K-1}+\frac{1}{n}\sum_{i\in I_b^k}(\nabla f_i(x^k) – y_i^{k-1})$, i.e., only using the updated minibatch. Also note that our ZeroSARAH method initializes all $y_i^{-1}$ variables  to be $0$ at the beginning (see Line 2 of our Algorithm 2).
>
> **Q3.** Regarding the reference [Inexact SARAH Algorithm for Stochastic Optimization] and the confusing/misleading statements.
>
> **A3.**
>
> - First, we emphasize that our **ZeroSARAH method is the first VR method that does not  need to compute any full gradients without assuming any additional assumptions (such as bounded gradient/variance), and also achieves the state-of-the-art convergence results.**
>
> - Re the reference Inexact SARAH: similarly to the SCSG/SVRG+/PAGE methods that we already discussed in our work, i.e., it avoids full gradient computations by using a large minibatch of stochastic gradients under the *bounded gradient/variance assumption*. Moreover, according to its Theorem 2 and Corollary 3 (nonconvex case), it does not match the best-known results given by e.g., SPIDER/PAGE and our ZeroSARAH.
>
> - For the misleading statement **“(our methods) improve the previous best-known result“** pointed by the reviewer, we do not think it is misleading since the complete sentence in our paper is **”(our methods) improve the previous best-known result (given by e.g., SPIDER, SARAH, and PAGE) in certain regimes.”** We point out that our results **do** improve previous best results for certain regimes, e.g., when $G_0$ is small,  which happens when the gradients at initial point $x^0$ are very close (please see our Table 1 and 2).
>
> **Q4.** Minor comments.
>
> **A4.** Thanks - we will take care of all minor comments and suggestions, and will make a thorough pass through the paper to make sure no possibility remains for any statements to be misleading or ambiguous.
>
> ---
>
> **We notice that the confidence of the reviewer is not reflected in the quality of the comments and criticism, as much of it is based on misunderstanding of our results. We hope that we managed to clarify these points. We would be most happy if this reviewer could re-evaluate our work in the light of our response.** We would wish to explicitly ask for justification.
>
> We are happy to respond to any further questions. We will do this as quickly as possible.
>
> Thank you!

---

> > ### Comment · Reviewer_bVt3 · 2021-11-25
> > **Author response**
> >
> > Thank you for your responses. I have discussed with other reviewers and my further comments are below.
> >
> > 1. Firstly, I do not misunderstand your results. In fact, the authors tried to misunderstand my comments. I only write the indices of $y_i^k$ for clarity in my review. I never say that we need to store all the variables for all $i$ and all $k$ every time in the algorithm. Instead of that, I wrote: "this algorithm needs to keep a table of past gradients in the auxiliary variables, similar to SAG/SAGA". This table has size $n \times d$ and it is expensive to store this when $n$ is large. This weakness is similar to SAG/SAGA and almost everyone in this area is familiar with this fact. I believe the authors know this too. However, they neither cited the previous works, nor discuss this fact in the original draft.
> >
> > 2. Secondly, when the authors introduce a new algorithm, it is important to discuss every elements in it and how to implement it. In Section 4.1, the description of your algorithm ZeroSARAH, the authors did not mention anything about the additional variables. Although these variables play an important role in their algorithm, that help eliminate the full gradient computations, the authors never discuss this fact in their original manuscript. They already know that "it is a trade-off (memory cost vs. no full gradient) for the proposed method". All I ask is that the authors clarify their weakness and properly discuss their implementation with existing literatures. However, they completely ignored my concerns and refused to improve their paper based on my suggestions.
> >
> > 3. Finally, regarding your statements:
> > - "our results can improve existing ones in certain regimes" is indeed misleading. The setting where there is improvement is restricted. As Reviewer qPn9 pointed out, "$G_0$ must be small and independent of $n$, hence it is not clear if it is possible to give a fair comparison between the current complexity and previous best complexity". In addition, when $ n \leq \mathcal{O}(\epsilon^{-4})$, there is no improvement in terms of complexity for your results. The authors failed to discuss this in the original draft. Note that it is important to consider the asymptotic behavior when $\epsilon \to 0$.  Intuitively, I don't think we can claim that the algorithm works well when the number of iteration is smaller than $\sqrt{n}$, i.e. when it did not gather enough information from $n$ individual functions.
> > -"all existing variance-reduced methods, including SARAH, SVRG, SAGA and their variants, need to compute the full gradient". Here are the full sentences in the abstract:
> >  "To the best of our knowledge, in this non-convex finite-sum regime, all existing variance-reduced methods, including SARAH,
> > SVRG, SAGA and their variants, need to compute the full gradient over all $n$ data samples at the initial point $x_0$ and then periodically compute the full gradient once every few iterations (for SVRG, SARAH and their variants)."
> > and
> > "The proposed ZeroSARAH and its distributed variant D-ZeroSARAH are the first variance-reduced algorithms which do not require any full gradient computations, not even for the initial point."
> > In these sentences, the authors do not mention any additional assumption/SOTA results that differentiate their method. I'm not sure if they remain correct.
> >
> > For these reasons, and because we can not see any changes of the paper during the revision period, my concerns remain the same.

---

> > > ### Author Response · Authors · 2021-11-26
> > > **To Reviewer bVt3**
> > >
> > > Thanks for your feedback. **However, we do not agree with your opinions above.**
> > >
> > > - For your Point 1.
> > > You said that *"I never say that we need to store all the variables for all $i$ and all $k$ every time in the algorithm."*. However, you indeed said this in your initial comments: *"ZeroSARAH needs a group of auxiliary variables $y_i^k$ for $i=1,\ldots, n$ and $k=0,1,2,\ldots$"*. **We do not want to argue about this point, people can judge by themselves.**
> > >
> > > - For your Point 2.
> > > The algorithm steps are clearly stated in Algorithm 2. Moreover, previous classical SAG/SAGA algorithms also directly write the summation of these $n$ variables $\sum_{j=1}^ny_j^k$ same as ours.  We do not think this point can be an issue/concern pointed by the reviewer (who wants to compute this sum explicitly). **In any case, this is just a minor point that does not affect any contributions of our work.**
> > >
> > > - For your Point 3.
> > > We just want to emphasize one fact: **there exist some regimes (no matter how small they are), our results can outperform previous results.** Also please see our response to Reviewer qPn9: **"Moreover, we would simply refer people to our Corollary 1 (which just computes the full gradient once at the initial point, while previous algorithms need to compute full gradients periodically) if you are very worried about $G_0$ or $\hat{\Delta}_0$ since in this case these terms do not appear in the final result!"**.
> > >   For your last sentence: *"In these sentences, the authors do not mention any additional assumption/SOTA results that differentiate their method. I'm not sure if they remain correct."*, **if you do not believe it, please just read the detailed proofs (also we want to point out that we have already discussed the additional assumptions e.g., bounded gradient/variance, and compared the results of previous algorithms in our Introduction Section and Tables).** According to your view of reviewing, you can directly reject any papers by saying "this paper may be wrong because I am not sure about their results".
> > >
> > > **In sum, we do not believe any of your concerns are indeed valid/exist and affect our contributions.**
> > >
> > > Authors

---

### Author Response · Authors · 2021-11-23
**To all reviewers**

We thank all reviewers for your time, effort, and for providing comments and suggestions.

---

We are fully convinced that no concern raised was major, and hence we do not think the low scores given by some reviewers are justified. The key concerns (apart from adding some clarifications, fine typos and so on) are based on a  **misunderstanding** by the reviewers. We have given detailed point-by-point responses to all reviewers for all concerns raised, and we hope our responses are helpful for the reviewers to understand our work more clearly. We hope that the reviewers will reevaluate our work. Thank you very much in advance!

---

We have received 4 scores: 8, 6, 5 and 3.  Thanks to Reviewers YmLP (score 8) and xTXr (score 6) who suggest acceptance. Reviewer qPn9 (score 5) said

> "Overall, I agree that an algorithm with zero full gradient computation is definitely interesting, it is not clear if this brings any advantage in theory (due to different parameters in the bounds) and also in practice (experiments are only compared to SARAH and not with the SOTA solvers in distributed settings.)"

and

> "I am of course open to increase my score if my concerns in 1, 2, 3 are addressed."

We believe we addressed these concerns, and hope that Reviewer qPn9 will increase their score. If we did not fully address any of these concerns, please do let us know, and we will make all necessary changes.
We thus fully expect 3 of the 4 reviewers to be supportive of acceptance.

We wish to highlight that while Reviewer bVt3 (score 3) self-identifies as "absolutely certain", his/her review is based on a number of factual errors and misunderstandings, which we clarify in our rebuttal. We would wish to kindly request all reviewers to read the review, and our response, as this will undoubtedly facilitate the post-rebuttal discussion among the reviewers and the AC. Please note that while we point this review out as the most inappropriate one (highest confidence, lowest score, serious factual errors in the review), we are still thankful to the reviewer for their hard work and effort! We are not complaining about the reviewer, we are merely complaining about the review. We hope Reviewer bVt3 will be able to engage with our response.

---

Finally, we wish to emphasize the key motivation/contribution of our work here, as perhaps some if it was misunderstood by some reviewers:


> Full gradient computations are time-consuming steps in many applications as the number of data samples $n$ usually is very large. Especially in the distributed setting, periodic computation of full gradient needs to periodically synchronize all clients/devices, which usually is impractical (e.g., partial participation is a key feature/requirement for federated learning). Motivated by this, we focus on designing new algorithms which do not require any full gradient computations. In particular, we propose the *first* variance-reduced algorithm ZeroSARAH (and also its distributed variant D-ZeroSARAH) *without computing any full gradients* for solving both standard and distributed nonconvex finite-sum problems. Moreover, ZeroSARAH and Distributed D-ZeroSARAH can obtain new state-of-the-art convergence results which improve previous best-known results (given by e.g., SPIDER, SARAH and PAGE) in certain regimes. Thus, we expect that ZeroSARAH/D-ZeroSARAH will have a practical impact in distributed and federated learning where full device participation is impractical. *Actually, we have already successfully applied ZeroSARAH into our following federated learning projects and achieved the desired results.*



**In the light of the above, and our detailed response below, we kindly ask the reviewers to reevaluate our work and consider increasing their scores if they believe we addressed their concerns. Thank you very much, wherever the discussion and decisions will lead!**

Best regards,

Authors

---

### Decision · Program_Chairs · 2022-01-20

**Decision:**

Reject

**Comment:**

There are many discussions among the reviewers for this paper and eventually none of the reviewers (including the one who gave most positive score) would like to support the publication of this paper.

Some concerns from the reviewers are as follows:
1. Missing the discussion on storage cost.
2. The improvement is limited. $G_0$ must be small and independent of $n$, hence it is not clear if it is possible to give a fair comparison between the current complexity and previous best complexity.
3. Missing the discussion on the case when $n \leq \mathcal{O}(\varepsilon^{-4})$ of the state-the-art results.
4. For the complexity results in terms of $\varepsilon$, it requires $\varepsilon$ to be arbitrarily small. The authors should also discuss this point for comparing with their result.
5. Some other statements in the papers are overclaimed.

Please take the comments and suggestions from the reviewers carefully to revise the paper for the future venues since they raised valid points.